# Calcium specificity signaling mechanisms in abscisic acid signal transduction in *Arabidopsis* guard cells

**Benjamin Brandt[†‡], Shintaro Munemasa[†§], Cun Wang, Desiree Nguyen, Taiming Yong, Paul G Yang, Elly Poretsky, Thomas F Belknap, Rainer Waadt, Fernando Alemán, Julian I Schroeder***

Division of Biological Sciences, Cell and Developmental Biology Section, University of California, San Diego, San Diego, United States

**\*For correspondence:**
jischroeder@ucsd.edu

[†]These authors contributed equally to this work

**Present address:** [‡]Structural Plant Biology Laboratory, Department for Botany and Plant Biology, University of Geneva, Geneva, Switzerland; [§]Graduate School of Environmental and Life Science, Okayama University, Okayama, Japan

**Competing interests:** The authors declare that no competing interests exist.

**Abstract** A central question is how specificity in cellular responses to the eukaryotic second messenger $Ca^{2+}$ is achieved. Plant guard cells, that form stomatal pores for gas exchange, provide a powerful system for in depth investigation of $Ca^{2+}$-signaling specificity in plants. In intact guard cells, abscisic acid (ABA) enhances (primes) the $Ca^{2+}$-sensitivity of downstream signaling events that result in activation of S-type anion channels during stomatal closure, providing a specificity mechanism in $Ca^{2+}$-signaling. However, the underlying genetic and biochemical mechanisms remain unknown. Here we show impairment of ABA signal transduction in stomata of calcium-dependent protein kinase quadruple mutant plants. Interestingly, protein phosphatase 2Cs prevent non-specific $Ca^{2+}$-signaling. Moreover, we demonstrate an unexpected interdependence of the $Ca^{2+}$-dependent and $Ca^{2+}$-independent ABA-signaling branches and the *in planta* requirement of simultaneous phosphorylation at two key phosphorylation sites in SLAC1. We identify novel mechanisms ensuring specificity and robustness within stomatal $Ca^{2+}$-signaling on a cellular, genetic, and biochemical level.

## Introduction

Cytosolic calcium ($[Ca^{2+}]_{cyt}$) functions as key cellular second messenger in a plethora of crucial processes in plants and other eukaryotes (*Hetherington and Woodward, 2003*; *Clapham, 2007*; *McAinsh and Pittman, 2009*; *Berridge, 2012*; *Charpentier and Oldroyd, 2013*; *Webb, 2013*). Elucidation of the mechanisms mediating specificity in $Ca^{2+}$ signaling is fundamental to understanding signal transduction (*Berridge et al., 2003*; *Hetherington and Woodward, 2003*; *Clapham, 2007*; *Webb, 2013*). In a few cases, the biochemical and cellular mechanisms mediating $Ca^{2+}$ signaling specificity have been revealed (e.g. *De Koninck and Schulman, 1998*; *Dolmetsch et al., 1998*; *Oancea and Meyer, 1998*; *Dolmetsch et al., 2001*; *Bradshaw et al., 2003*; *Rellos et al., 2010*; *Chao et al., 2011*). More than one (non-exclusive) mechanism is likely to contribute to specificity in $Ca^{2+}$ signal transduction (*Berridge et al., 2003*; *Dodd et al., 2010*). However, characterization of the combined cellular, biochemical, and genetic mechanisms underlying $Ca^{2+}$ specificity in a single cell type has not been achieved to our knowledge.

The genome of the plant *Arabidopsis thaliana* encodes over 200 EF-hand $Ca^{2+}$-binding proteins (*Day et al., 2002*), with many of these genes co-expressed in the same cell types (*Harmon et al., 2000*; *McCormack et al., 2005*; *Schmid et al., 2005*; *Winter et al., 2007*), illustrating the need for $Ca^{2+}$ specificity signaling mechanisms in plants. Two guard cells form a stomatal pore representing the gateway for $CO_2$ influx, which is inevitably accompanied by plant water loss. The aperture of stomatal pores is consequently tightly regulated by the guard cells. Intracellular $Ca^{2+}$ represents a key second messenger in stomatal closing (*McAinsh et al., 1990*; *MacRobbie, 2000*; *Hetherington, 2001*; *Hetherington and Woodward, 2003*; *Hubbard et al., 2012*), but intracellular $Ca^{2+}$ also functions in

**eLife digest** Plant leaves have tiny openings or pores called stomata, which allow carbon dioxide, water vapor and other gases to diffuse in and out of the plant. Two cells called guard cells surround each stoma and control the opening and closing of the pore. If a plant is losing excessive amounts of water, for example during a drought, the plant produces a hormone called abscisic acid that promotes the closure of its stomata.

When abscisic acid is present, the guard cells are sensitive to changes in their internal concentration of calcium ions so that calcium ions can activate a protein called SLAC1. This leads to responses in the guard cells that close the stoma. The calcium ions activate SLAC1 by stimulating enzymes called calcium-dependent protein kinases (CPKs). However, abscisic acid can also trigger other enzymes that can activate SLAC1 independently of the calcium ions.

Calcium ions are also reported to be involved in the opening of stomata, when abscisic acid is not present. Therefore, it is not clear how abscisic acid works to specifically 'prime' guard cells to close the stomata in response to increases in calcium ions during drought. Brandt, Munemasa et al. studied stomata in a plant called *Arabidopsis thaliana*. The experiments show that, in the presence of abscisic acid, mutant plants that lack four different CPK enzymes are impaired in the activation of SLAC1 and the closing of stomata in response to increases in calcium ions.

Further experiments found that other enzymes called the PP2Cs—which are switched off by abscisic acid—are responsible for regulating the $Ca^{2+}$ sensitivity of guard cells. Switching off PP2Cs enables closing of the stomata in response to calcium ions. It has been suggested previously that the CPKs and the calcium-independent enzymes are involved in two separate pathways that promote the closure of stomata. However, Brandt, Munemasa et al. found that the calcium-independent enzymes are required for calcium ions to activate SLAC1 in guard cells, revealing that these two pathways are linked.

Brandt, Munemasa et al.'s findings reveal how abscisic acid is able to specifically prime guard cells to close stomata in response to calcium ions. The next challenge is to understand how the CPKs and calcium-independent enzymes work together during the closure of stomata.

stomatal opening (*Irving et al., 1992*; *Shimazaki et al., 1992*; *Curvetto et al., 1994*; *Shimazaki et al., 1997*; *Cousson and Vavasseur, 1998*; *Young et al., 2006*), raising the question how cytosolic free $Ca^{2+}$ concentration ($[Ca^{2+}]_{cyt}$) elevations trigger a specific cellular response. The underlying mechanisms mediating specificity in guard cell $Ca^{2+}$ signaling are not well understood. The development of genetic, electrophysiological, and cell signaling tools for the dissection of $Ca^{2+}$ signaling within this model cell type renders guard cells a powerful system for the investigation of specificity mechanisms within $Ca^{2+}$ signal transduction. Recent studies including analyses in intact *Arabidopsis* (*Young et al., 2006*) and *Vicia faba* (*Chen et al., 2010*) guard cells, have shown that stomatal closing stimuli including abscisic acid (ABA) and $CO_2$ enhance the $[Ca^{2+}]_{cyt}$ sensitivity of downstream signaling mechanisms, switching them from an inactivated state to an enhanced $Ca^{2+}$-responsive 'primed' state, thus tightly controlling specificity in $Ca^{2+}$ responsiveness (*Young et al., 2006*; *Munemasa et al., 2007*; *Siegel et al., 2009*; *Chen et al., 2010*; *Xue et al., 2011*). A rise of $[Ca^{2+}]_{cyt}$ from resting to elevated levels alone does not trigger the full ion channel regulation and stomatal response (*Young et al., 2006*; *Munemasa et al., 2007*; *Siegel et al., 2009*; *Chen et al., 2010*; *Xue et al., 2011*). Similarly, a recent study of pathogen-associated molecular pattern (PAMP) signaling suggests that prior PAMP signaling enhances the sensitivity to intracellular $Ca^{2+}$ during signal transduction (*Kadota et al., 2014*), indicating that this principle for $Ca^{2+}$ specificity priming may be more widely used in plants. The biological closing stimulus has to be present for the guard cell to react to physiological $Ca^{2+}$ elevation. However, the biochemical and genetic mechanisms mediating $Ca^{2+}$ sensitivity priming remain unknown.

SLAC1 represents the major anion channel mediating S-type anion currents in guard cells (*Negi et al., 2008*; *Vahisalu et al., 2008*) and $Ca^{2+}$ activation of S-type anion currents is an early and crucial step in stomatal closure (*Schroeder and Hagiwara, 1989*; *McAinsh et al., 1990*; *Siegel et al., 2009*; *Chen et al., 2010*). $Ca^{2+}$-independent SnRK2 protein kinases (*Li et al., 2000*; *Mustilli et al., 2002*; *Yoshida et al., 2002*), most importantly OST1, have been shown to activate SLAC1 in *Xenopus leavis* oocytes (*Geiger et al., 2009*; *Lee et al., 2009*; *Brandt et al., 2012*). The full length $Ca^{2+}$-dependent

protein kinases 6, 21, and 23 (CPK6, CPK21, and CPK23) also activate SLAC1 in oocytes (*Geiger et al., 2010*; *Brandt et al., 2012*). Presently, the $Ca^{2+}$-dependent and $Ca^{2+}$–independent branches are considered to function independently (e.g. *Li et al., 2006*; *Kim and et al., 2010*; *Roelfsema et al., 2012*). The activation of SLAC1 by OST1 or CPK6 is inhibited by the clade A protein phosphatase 2Cs (PP2Cs) ABI1, ABI2, or PP2CA in oocytes (*Geiger et al., 2009*; *Lee et al., 2009*; *Brandt et al., 2012*). The cytosolic ABA-receptors pyrabactin resistance (PYR)/PYR-like (PYL)/regulatory component of ABA receptor (RCAR) (*Ma et al., 2009*; *Park and et al., 2009*) have been shown to inhibit PP2C activity in the presence of ABA (*Ma et al., 2009*; *Park and et al., 2009*; *Santiago et al., 2009*; *Nishimura et al., 2010*; *Szostkiewicz et al., 2010*). Reconstitution of ABA activation of SLAC1 in *Xenopus* oocytes has been shown by co-expression of the ABA-receptor PYR1 together with SLAC1, PP2Cs, and either $Ca^{2+}$-independent OST1 or $Ca^{2+}$-dependent CPK6 protein kinases (*Brandt et al., 2012*). However, whether the $Ca^{2+}$-dependent and–independent branches in ABA signal transduction are functionally linked and depend on one-another *in planta* remains to be investigated using higher order genetic mutants. Here we present biochemical, genetic and cellular signaling findings that describe mechanisms underlying specificity and robustness in $Ca^{2+}$ signaling within a single cell type and demonstrate an unexpected strong dependence of the $Ca^{2+}$-dependent signal transduction branch on the $Ca^{2+}$-independent pathway in guard cells. Moreover our results suggest that in contrast to OST1 (*Umezawa et al., 2009*; *Vlad et al., 2009*), calcium-dependent protein kinases (CPKs) are not directly deactivated by PP2Cs, but these PP2Cs rapidly deactivate both of the $Ca^{2+}$-dependent and $Ca^{2+}$–independent branches by directly dephosphorylating the protein kinase target SLAC1.

## Results

### CPK requirement for ABA activation of anion channels

Previous studies have shown that *A. thaliana* single or double mutants in CPKs cause partial ABA-insensitivities in guard cell signaling (*Mori et al., 2006*; *Zhu et al., 2007*; *Hubbard et al., 2012*). We addressed the question whether higher order *CPK* gene disruption mutant plants display more strongly impaired ABA responses. CPK6 and CPK23 were shown to activate SLAC1 in *Xenopus* oocytes and disruption of the corresponding genes in plants leads to a partial reduction of S-type anion current activation in guard cells (*Mori et al., 2006*; *Geiger et al., 2010*; *Brandt et al., 2012*). The closest homolog to CPK6, CPK5, is associated with reactive oxygen species signaling (*Boudsocq et al., 2010*; *Dubiella et al., 2013*). CPK5 also activates SLAC1 in oocytes (*Figure 1—figure supplement 1A,B*). Whole-cell patch-clamp analysis showed that mutation of *CPK5* alone does not substantially disrupt ABA-activation of S-type anion channels (*Figure 1—figure supplement 1C,D*), consistent with findings of over-lapping gene functions in this response (*Mori et al., 2006*; *Hubbard et al., 2012*). CPK11 is highly expressed in guard cells and involved in ABA responses (*Zhu et al., 2007*; *Geiger et al., 2009*). We isolated *cpk5/6/11/23* quadruple T-DNA insertion mutant plants and investigated ABA-induced S-type anion channel current regulation. Either ABA treatment (*Siegel et al., 2009*) or by-passing ABA signaling by exposure of guard cells to a high external $Ca^{2+}$ shock (*Allen et al., 2002*) renders wildtype (Col0) guard cells sensitive to physiological $[Ca^{2+}]_{cyt}$ increases. Notably, even when previously exposed to ABA or a high external $Ca^{2+}$ shock, 2 μM $[Ca^{2+}]_{cyt}$ did not result in S-type anion current activation in *cpk5/6/11/23* quadruple mutant guard cells in contrast to WT plants (*Figure 1A–D*). These results show an important role of these calcium sensing protein kinases in ABA-dependent S-type anion channel activation in guard cells. We further investigated ABA-induced stomatal movement responses. Application of 5 μM ABA to WT leaves significantly decreased stomatal apertures compared to mock-treated control stomatal apertures (*Figure 1E*; p < 0.05). In the *cpk5/6/11/23* mutant, however, 5 μM ABA-induced stomatal closing was not significant (*Figure 1E*; p = 0.51). When the ABA concentration was increased to 10 μM, ABA-induced stomatal closure was weakened in *cpk5/6/11/23* mutant leaves (*Figure 1F*; p = 0.07; 0 min ABA-exposed *cpk5/6/11/23* mutant leaves compared to 60 min ABA-exposed *cpk5/6/11/23* mutant leaves). The partial ABA response at the higher ABA concentration may be linked to parallel activation of R-type anion channels (see 'Discussion').

### Constitutive $[Ca^{2+}]_{cyt}$ activation of S-type anion channels and primed $Ca^{2+}$-dependent stomatal closure in *pp2c* quadruple mutant guard cells

Members of the clade A of the PP2C class play important roles as negative regulators of ABA signaling (*Cutler et al., 2010*) and were shown to inhibit CPK-activation of SLAC1 in oocytes

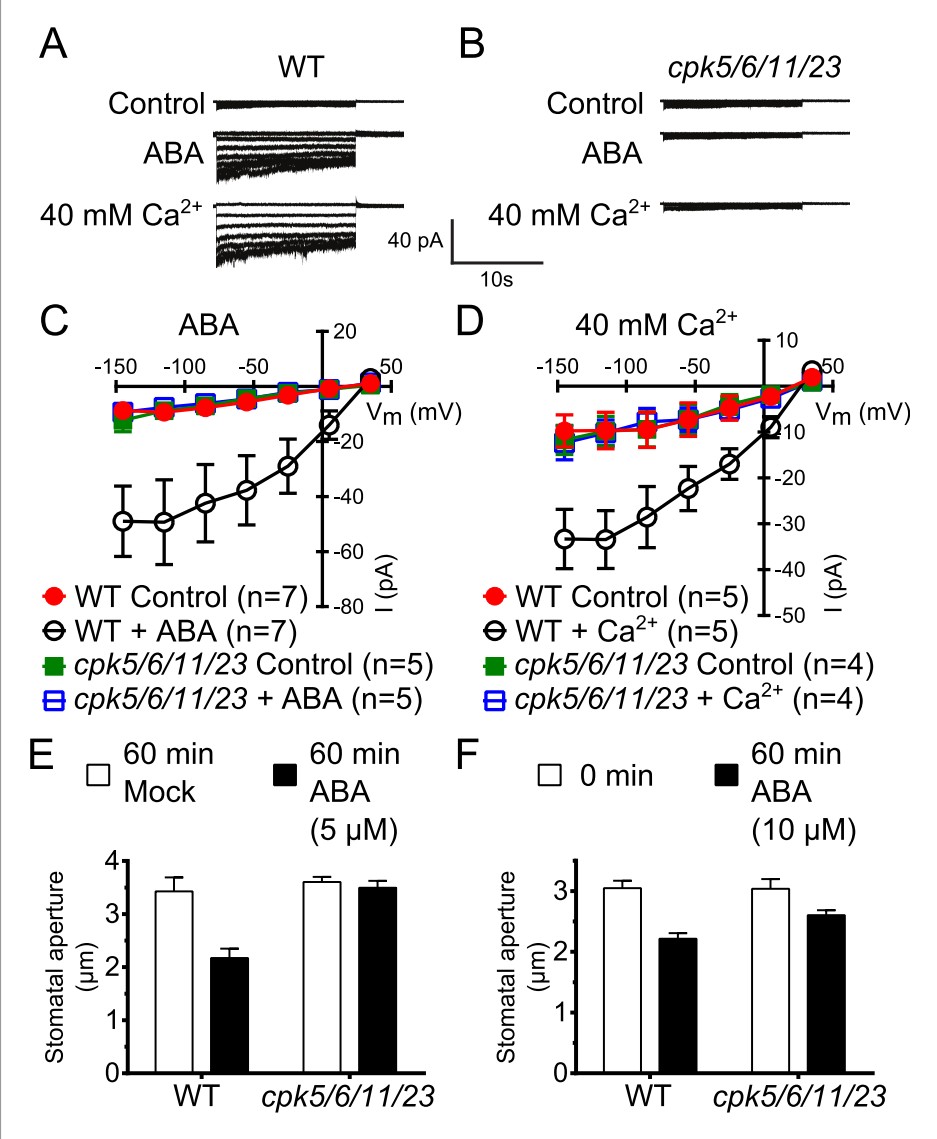

**Figure 1**. Calcium-dependent protein kinase (CPK) quadruple loss of function mutants show abscisic acid (ABA) and $Ca^{2+}$ insensitive S-type anion current activation and are impaired in stomatal closing. (**A**–**D**) Intracellular $Ca^{2+}$-activation of S-type anion channels enabled by pre-exposure to ABA (**A** and **C**) or high external $Ca^{2+}$ pre-shock (*Allen et al., 2002*) (**B** and **D**) is strongly impaired in *cpk5/6/11/23* guard cells at 2 µM $[Ca^{2+}]_{cyt}$. (**E** and **F**) 5 µM ABA-application to intact leaves shows impaired ABA-induced stomatal closing in *cpk5/6/11/23* mutant plants (**E**; p = 0.51 Mock-treated *cpk* quadruple mutant vs ABA-treated *cpk* quadruple mutant stomata; unpaired t-test; n = 6 experiments and $\geq$51 total stomata per group). Application of 10 µM ABA results in a partially reduced average stomatal response (**F**, p = 0.07; 0 min ABA-exposed *cpk5/6/11/23* mutant leaves compared to 60 min ABA-exposed *cpk5/6/11/23* mutant leaves; Student's *t*-test; n = 3 experiments and >59 total stomata per group). Representative whole cell currents (**A** and **B**), average steady-state current–voltage relationships ±SEM (**C** and **D**), average guard cell apertures ±SEM (**E** and **F**) are shown. Measurements shown in *Figure 1C* and *Figure 1—figure supplement 1D* were acquired under the same experimental condition. Therefore, WT Control and WT + ABA control data are the same in both figures. Several error bars are not visible, as these were smaller than the illustrated symbols.

The following figure supplement is available for figure 1:

**Figure supplement 1**. CPK5 activates SLAC1 in Xenopus oocytes and ABA-activation of S-type anion currents in *cpk5* single mutant is not impaired.

(*Geiger et al., 2010*; *Brandt et al., 2012*). To determine whether these PP2Cs function in the ABA-triggered enhancement of the $[Ca^{2+}]_{cyt}$-sensitivity in guard cells, we performed whole-cell patch-clamp analysis using a plant line carrying T-DNA insertion mutations in the key ABA signaling PP2Cs *ABI1*, *ABI2*, *HAB1*, and *PP2CA* (*abi1-2/abi2-2/hab1-1/pp2ca-1*). Surprisingly, in *abi1-2/abi2-2/hab1-1/pp2ca-1* quadruple mutant guard cells, strong $Ca^{2+}$-activated S-type anion currents were observed even without pre-exposure to ABA (*Figure 2A–D*). At low 0.1 µM $[Ca^{2+}]_{cyt}$ S-type anion channels did not show significant activation in the *pp2c* quadruple mutant compared to WT (*Figure 2—figure supplement 1A,B*; p = 0.294 at −145 mV). These findings provide genetic evidence for first genes that are essential for the ABA-triggered $Ca^{2+}$ sensitivity priming in guard cells and show that these PP2Cs provide a mechanism ensuring specificity in $Ca^{2+}$ signal transduction.

## CPK activities are not directly ABA-regulated and disruption of PP2Cs does not cause constitutive activation of OST1

Based on the above results we sought to determine the biochemical mechanisms by which PP2Cs down-regulate $Ca^{2+}$ sensitivity in the absence of ABA. The main SLAC1-activating protein kinase in the $Ca^{2+}$-independent branch, OST1 (*Mustilli et al., 2002*; *Yoshida et al., 2002*), is directly inactivated by PP2Cs through de-phosphorylation of the activation loop (*Umezawa et al., 2009*; *Vlad et al., 2009*).

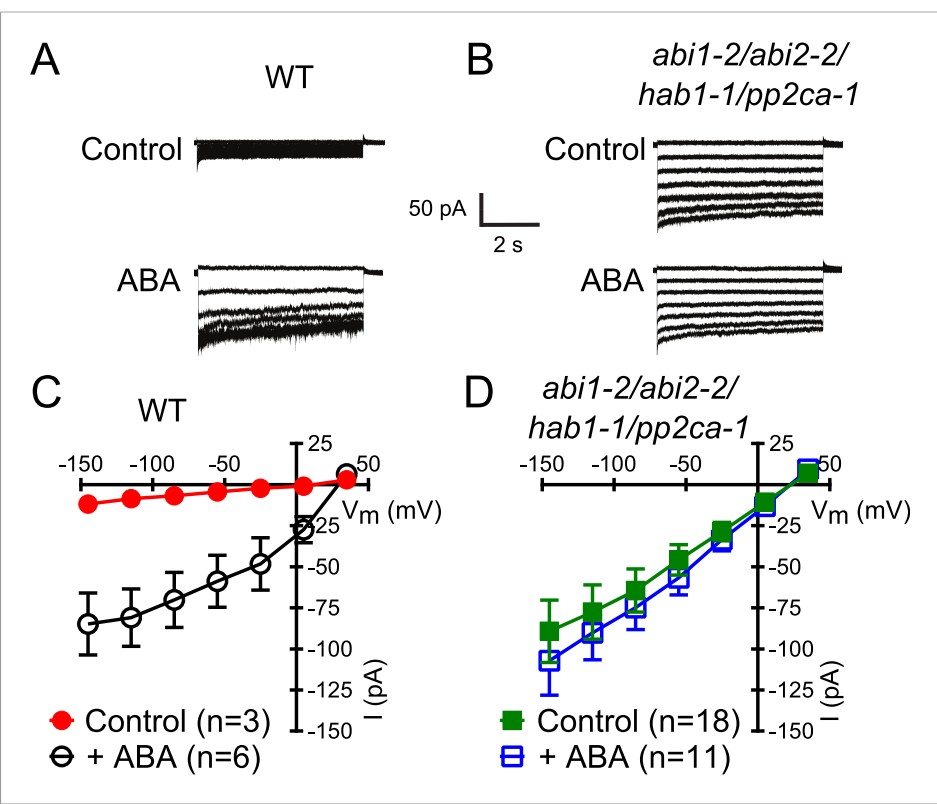

**Figure 2**. In protein phosphatase 2C (PP2C) quadruple mutant plants, $Ca^{2+}$ activation of S-type anion currents is constitutively primed. (**A** and **C**) 2 µM $[Ca^{2+}]_{cyt}$ activates S-type anion currents in WT if the guard cells were pre-exposed to ABA. (**B** and **D**) In PP2C quadruple mutant guard cells ABA pre-exposure is not required for 2 µM $[Ca^{2+}]_{cyt}$-activation of S-type anion currents. Average steady-state current–voltage relationships ±SEM, guard cell numbers (**C** and **D**), and representative whole cell currents (**A** and **B**) are presented. Several error bars are not visible, as these were smaller than the illustrated symbols.

The following figure supplement is available for figure 2:

**Figure supplement 1**. Analysis of ABA activation of S-type anion currents in PP2C quadruple mutant guard cells at low $[Ca^{2+}]_{cyt}$.

We tested whether CPKs might be down-regulated by PP2Cs in a similar manner and whether *pp2c* quadruple mutant plants may also exhibit a constitutive OST1 activity. Our first approach to test whether CPK activity is regulated by ABA through PP2Cs was an in-gel protein kinase assay using protein extracts of *Arabidopsis* seedlings, which is routinely used to test OST1 activation by ABA (*Mustilli et al., 2002*) and also CPK activation by flg22 (*Boudsocq et al., 2010*). Guard cell $[Ca^{2+}]_{cyt}$ ranges from resting levels of $\approx 0.15$ μM to stimulus induced elevated levels of above 1 μM (*McAinsh et al., 1990*). Similar to studies reporting the ABA-activation of SnRK2.2, SnRK2.3, and SnRK2.6/OST1 (*Mustilli et al., 2002*; *Yoshida et al., 2002*; *Fujii et al., 2007*), we compared the phosphorylation pattern of a reaction carried out at 0.15 μM free $Ca^{2+}$ with the phosphorylation pattern at 3 μM free $Ca^{2+}$ (*Figure 3A,B*; for intermediate free $Ca^{2+}$ concentration of 0.4 μM $Ca^{2+}$ see *Figure 3—figure supplement 2*). Incubating the gels in a reaction buffer with 3 μM free $Ca^{2+}$ led to strong $Ca^{2+}$-activated phosphorylation signals compared to resting $Ca^{2+}$ at 0.15 μM (*Figure 3A,B*). To determine whether these $Ca^{2+}$-activated signals are CPK-derived we included two distinct quadruple mutants, *cpk5/6/11/23* and *cpk1/2/5/6*, in the in-gel kinase assays. Several $Ca^{2+}$-activated bands disappeared or became notably weaker when extracts were tested from *cpk5/6/11/23* and *cpk1/2/5/6* (*Boudsocq et al., 2010*) plants (*Figure 3B* and for improved visibility *Figure 3—figure supplement 1*).

Exposure of *Arabidopsis* seedlings to ABA led to OST1 protein kinase activation, confirming functional ABA responses (*Figure 3A,B*, lanes 1–2 and 9–10; 'OST1' inset). However, CPK-derived band intensities did not change in the presence of ABA, indicating that CPK activities may not be directly ABA-regulated, in contrast to OST1 (*Figure 3B*). These findings were also obtained at an intermediate free $Ca^{2+}$ concentration of 0.4 μM (*Figure 3—figure supplement 2A,B*). Moreover, in-gel CPK protein kinase activities were not altered with or without ABA in seedling extracts of *abi1-2/abi2-2/hab1-1/ pp2ca-1* quadruple mutant plants (*Figure 3A,B*, lanes 3–4 and 11–12; *Figure 3—figure supplement 1A,B*). Interestingly, the *pp2c* quadruple mutants did not enable constitutive OST1 activation in vivo, differing from (*Fujii et al., 2009*), but consistent with (*Vlad et al., 2009*) (*Figure 3A,B*, lanes 3–4 and 11–12 and *Figure 3—figure supplement 2A,B*; see 'OST1' inset). Furthermore, OST1-derived band intensities were not changed in the *cpk5/6/11/23* and *cpk1/2/5/6* mutant plants showing that these *cpk* quadruple mutants retain ABA-activation of OST1 (*Figure 3A,B*, lanes 5–8 and 13–16; see 'OST1' inset).

## PP2Cs do not down-regulate CPK6 kinase activity directly

Initially, we tested whether the signals found in in-gel protein kinase assays are derived from kinase auto-phosphorylation or due to trans-phosphorylation activities of the protein kinases. To distinguish between auto- and trans-phosphorylation activities of recombinant CPK6 and OST1 we compared in-gel band intensities of gels with or without the substrate Histone-III (*Figure 3—figure supplement 3A,B*). The strong reduction of band intensities for recombinant CPK6 and OST1 when no Histone-III is present (*Figure 3—figure supplement 3A,B*) indicates that the signals observed in in-gel protein kinase assays are largely derived from CPK6 and OST1 kinase trans-phosphorylation activities of Histone-III consistent with previous reports for CPKs involved in pathogen signaling (*Boudsocq et al., 2010*).

To determine whether PP2Cs can directly down-regulate CPKs we next investigated whether the SLAC1-activating CPK6 (*Mori et al., 2006*; *Brandt et al., 2012*), is negatively regulated by the PP2Cs ABI1 and PP2CA. In-gel protein kinase assays using recombinant proteins were pursued in which kinases and phosphatases are separated by size prior to substrate phosphorylation. CPK6, and as positive control OST1, were pre-incubated either alone or with ABI1 or PP2CA with and without ATP before being subjected to in-gel protein kinase assays. Pre-incubation with either ABI1 or PP2CA did not inhibit CPK6 trans-phosphorylation activity (*Figure 3C*, lanes 2–3 and 5–6). In contrast, control OST1-derived substrate phosphorylation band intensities strongly decreased when ABI1 or PP2CA proteins were present during the pre-incubation period (*Figure 3D*, lanes 2–3 and 5–6). These results indicate that OST1, but not CPK6 activity, is directly down-regulated by ABI1 and PP2CA. CPKs have been previously reported to interact with ABI1 (*Geiger et al., 2010*). An electro-mobility shift can be observed for OST1 as well as for CPK6 (*Figure 3C,D*). These shifts could be due to dephosphorylation of CPK6 (*Figure 3—figure supplement 4*) and OST1 (*Umezawa et al., 2009*; *Vlad et al., 2009*) by PP2Cs. However, dephosphorylation by PP2Cs did not inhibit CPK6 activity (*Figure 3C*). An additional independent biochemical assay measuring ATP consumption also did not show down-regulation of

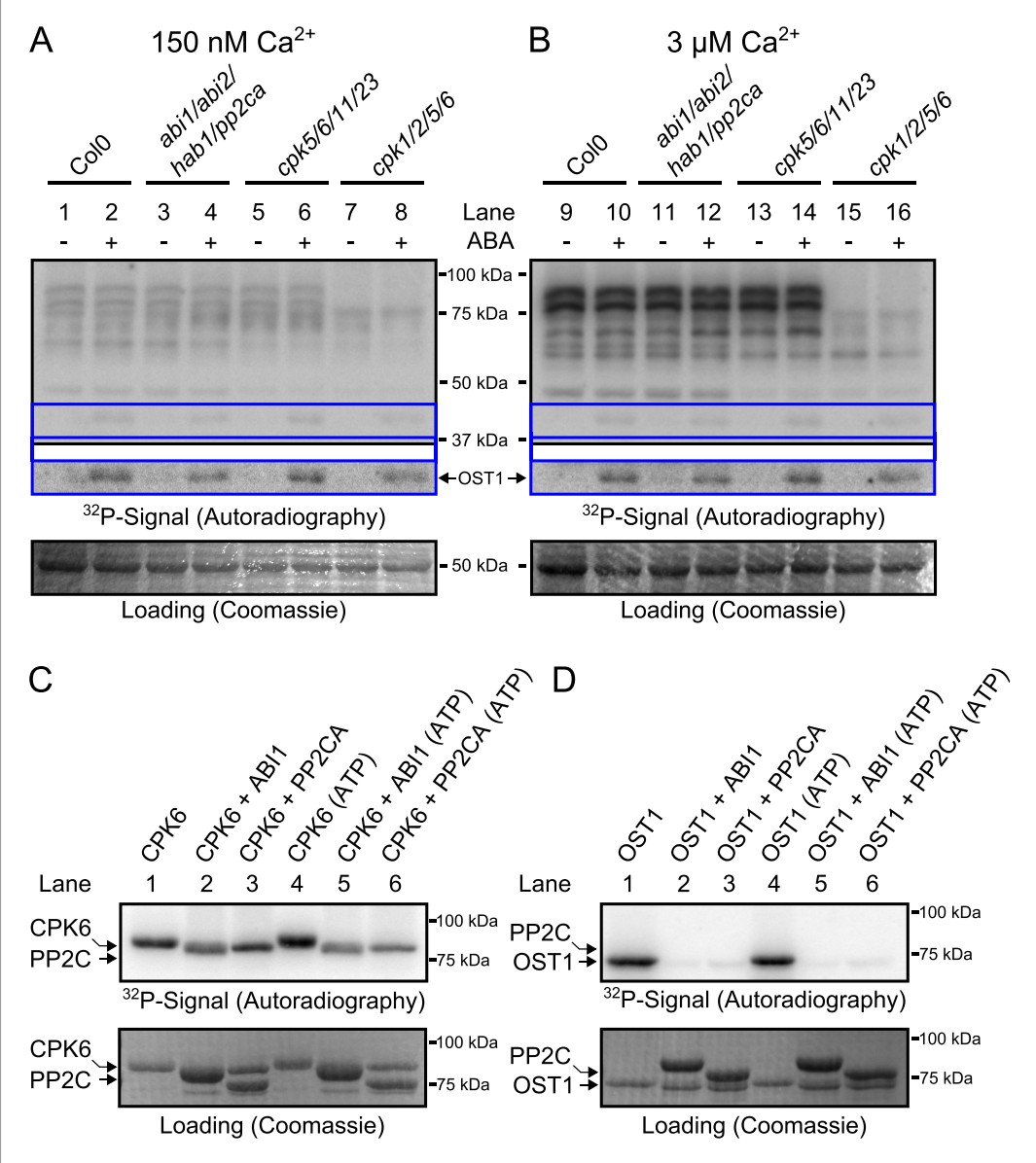

**Figure 3**. CPK activity is not changed by ABA or hyper-activated in *pp2c* quadruple mutants at defined $Ca^{2+}$ concentrations. (**A** and **B**) In-gel kinase assays with Histone-III as substrate for whole plant protein extracts show (**B**) 3 µM $Ca^{2+}$-activated trans-phosphorylation kinase activities independent of application of 50 µM ABA (lanes 9 and 10). In contrast, ABA activation of OST1 is clearly visible (lanes 1–2 and 9–10 at ~41 kDa; lower 'OST1' inset shows the same signal optimized autoradiography at the ~41 kDa region and the corresponding gel regions are indicated by blue lines; see 'Materials and methods'). Disruption of four PP2C genes (*ABI1*, *ABI2*, *HAB1*, and *PP2CA*) does not result in constitutive $Ca^{2+}$-activated and OST1 kinase activities (lanes 3–4 and 11–12). In-gel kinase activities of two independent CPK quadruple mutant lines indicate that the $Ca^{2+}$-activated kinase signals are CPK-derived (compare lanes 9–10 with 13–16 in **B** and see *Figure 3—figure supplement 1*); predicted MWs for CPK1, CPK2, CPK5, CPK6, CPK11, and CPK23 are 68.3 kDa, 72.3 kDa, 62.1 kDa, 61.1 kDa, 55.9 kDa, 58.7 kDa, respectively. (**C** and **D**) In-gel protein kinase assays with recombinant proteins show that incubation of the protein kinases with the PP2Cs ABI1 and PP2CA does (**C**) not change CPK6 activity while (**D**) OST1 activity is strongly down-regulated by PP2Cs. Each experiment has been repeated at least three times with similar results.

The following figure supplements are available for figure 3:

**Figure supplement 1**. Close up view of $Ca^{2+}$-activated kinase activities.

*Figure 3. continued on next page*

*Figure 3. Continued*

**Figure supplement 2**. Protein kinase activities are not altered by ABA-application at 150 nM and 400 nM free $Ca^{2+}$.

**Figure supplement 3**. Signals in in-gel kinase assays are largely derived from kinase trans-phosphorylation activities.

**Figure supplement 4**. CPK6 is de-phosphorylated by the PP2Cs ABI1, ABI2, and PP2CA.

**Figure supplement 5**. CPK6 kinase activity is not inhibited in the presence of ABI1 or PP2CA.

CPK6 activity in the presence of ABI1 and PP2CA (*Figure 3—figure supplement 5*), further underlining no direct down-regulation of CPK6 activity by these three PP2Cs, in contrast to OST1 controls.

## PP2Cs interact with and rapidly dephosphorylate SLAC1

Our results suggest that PP2Cs neither down-regulate CPK6 activity directly in vitro (*Figure 3C,D* and *Figure 3—figure supplement 5*) nor that CPK activities are strongly ABA-regulated independent of $[Ca^{2+}]$ changes in native plant protein extracts (*Figure 3A,B*). We next investigated the kinetics and specificity of PP2C down-regulation of SLAC1 activation by CPKs through dephosphorylation of the SLAC1 channel, a mechanism reported for CPK-dependent transcription factor regulation (*Lynch et al., 2012*) and consistent with previous findings (*Brandt et al., 2012*). First, we determined whether SLAC1 interacts with the PP2C ABI1 *in planta* using bimolecular fluorescence complementation (BiFC). We observed clear BiFC signals for full length SLAC1 co-expressed with CPK6 and ABI1 (*Figure 4A,B*) while signal intensities of SLAC1 co-expressed with a control protein phosphatase 2A catalytic subunit 5 (PP2AC5) were very low (*Figure 4B*). Protein–protein interaction of SLAC1 with PP2CA in BiFC experiments was reported earlier (*Lee et al., 2009*). As shown in *Figure 4C,D*, the ABI1-mediated dephosphorylation of the N-terminus of SLAC1 (SLAC1-NT) previously phosphorylated by CPK6 (*Brandt et al., 2012*) occurs very rapidly. Already 1 min after the addition of ABI1 a strong decrease of the phosphorylation signal was observed (*Figure 4D*, lane 4). This de-phosphorylation was also found when the PP2C phosphatase PP2CA was added instead of ABI1 (*Figure 4C,E*, lane 4). To test whether this is a general phenomenon, we phosphorylated the SLAC1-NT with the SLAC1-activating and -phosphorylating kinases CPK21, CPK23, and OST1 (*Geiger et al., 2009, 2010*; *Lee et al., 2009*) and analyzed whether ABI1 and PP2CA are able to remove phospho-groups added by these kinases (*Figure 4F–H* and *Figure 4—figure supplement 1*). After inhibiting the kinase with staurosporine, band intensities decreased only after addition of the PP2C protein phosphatases for all combinations, showing that this rapid SLAC1 de-phosphorylation is mediated by PP2Cs (*Figure 4F–H* and *Figure 4—figure supplement 1*, lanes 5–6).

## Disruption of $Ca^{2+}$-independent SnRK kinases impairs $Ca^{2+}$-dependent S-type anion channel regulation

The $Ca^{2+}$-independent and $Ca^{2+}$-dependent branches of ABA signal transduction are presently considered to be independent (e.g., *Li et al., 2006*; *Kim and et al., 2010*; *Roelfsema et al., 2012*), but this model has not been genetically investigated in *Arabidopsis*. In the *cpk5/6/11/23* quadruple mutant, ABA-activation of S-type anion currents and stomatal closure were impaired (*Figure 1A–E*), providing evidence for a possible interdependence of these signaling branches. The *ost1* single gene disruption mutant in the Col ecotype shows intermediate S-type anion current activation by ABA (*Geiger et al., 2009*). Three $Ca^{2+}$-independent SnRK kinases, SnRK2.2, SnRK2.3, and OST1 can activate SLAC1 in oocytes (*Geiger et al., 2009*) and redundantly function in controlling leaf water loss (*Fujii and Zhu, 2009*). Interestingly, *snrk2.2/snrk2.3/ost1* triple mutants were strongly impaired in ABA activation and notably also external $Ca^{2+}$ shock-induced activation of S-type anion channels at 2 μM $[Ca^{2+}]_{cyt}$ (*Figure 5A–D*). Imposing repetitive cytosolic $Ca^{2+}$ transients by alternating guard cell incubation buffers induces a fast $Ca^{2+}$-reactive stomatal closure response (*Allen et al., 2001*). We further analyzed imposed $Ca^{2+}$ oscillation-induced stomatal closure in *snrk2.2/snrk2.3/ost1* triple mutants. $Ca^{2+}$ reactive stomatal closure of the *snrk* triple mutant was impaired compared to wildtype plants (*Figure 5E*, p < 0.02 for wildtype vs *snrk2.2/snrk2.3/ost1* at 120 min). These data show that

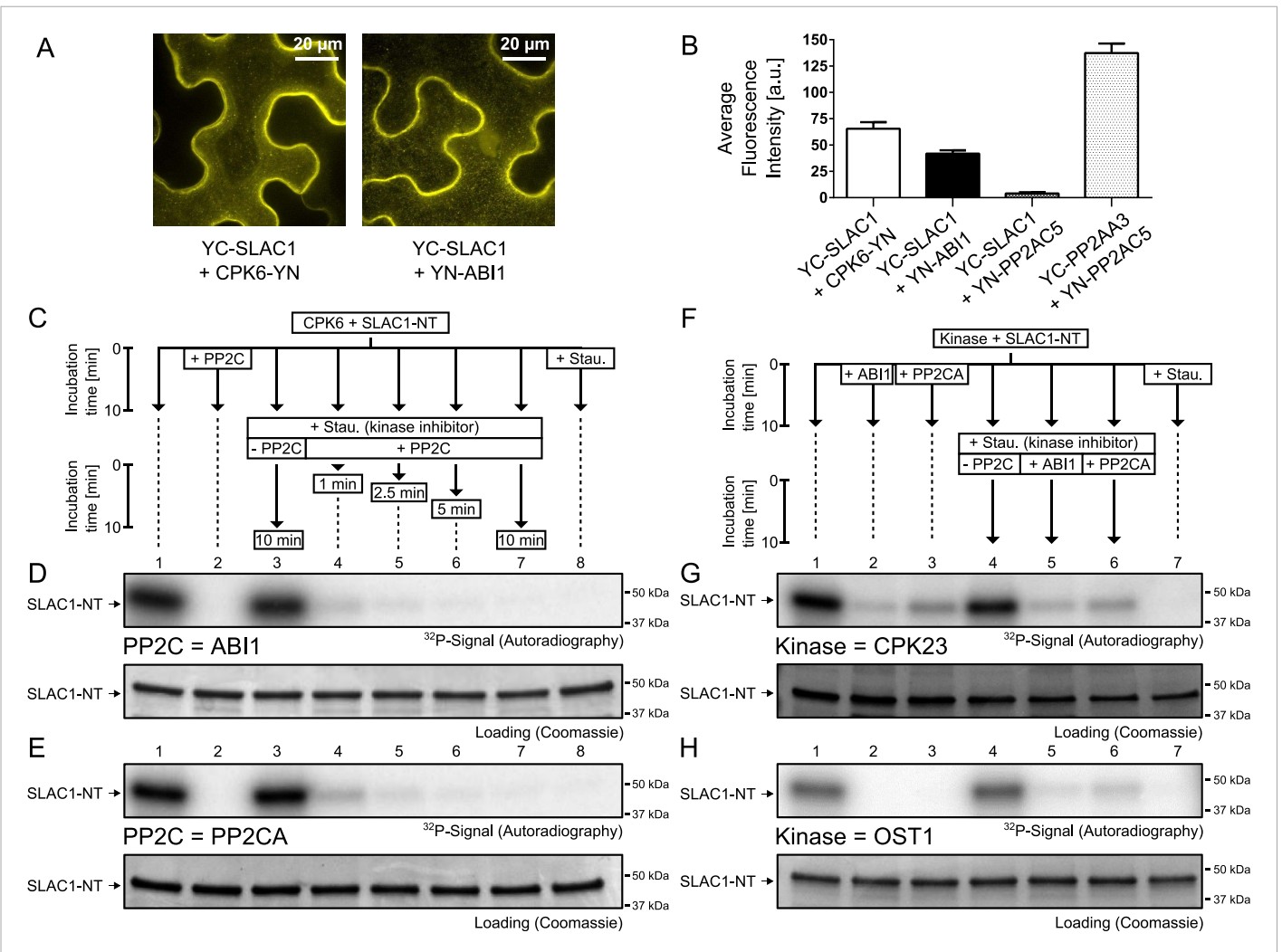

**Figure 4**. PP2Cs interact with and directly and rapidly dephosphorylate the N-terminus of SLAC1 (SLAC1-NT) when previously phosphorylated by several SLAC1-activating CPK and OST1 protein kinases. (**A**) Bimolecular fluorescence complementation (BiFC) experiments in *Nicotiana benthamiana* leaves show YFP-derived fluorescence signals of YC-SLAC1 co-expressed with CPK6-YN and YN-ABI1. (**B**) Quantification of BiFC-mediated YFP-fluorescence shows that SLAC1 interacts with CPK6 and ABI1 but not with the control catalytic protein phosphatase 2A subunit C5 (PP2AC5). YFP signals of positive control YN-PP2AC5 with protein phosphatase 2A regulatory subunit A3 fused to YC (YC-PP2AA3) confirm expression of PP2AC5. Data shown in (**B**) represent the average fluorescence intensity of randomly picked leaf areas (n = 40; ±SEM) and these data are also included in *Figure 6—figure supplement 5*. (**C–E**) CPK6-phosphorylated SLAC1-NT is rapidly de-phosphorylated by ABI1 and PP2CA. SLAC1-NT phosphorylation by CPK6 (**D** and **E**, lane 1) is strongly inhibited if the PP2C protein phosphatase was added before starting the reaction (**D** and **E**, lane 2), but remains stable after addition of elution buffer (Elu.) and kinase inhibitor staurosporine (Stau.) with subsequent 10 min incubation (**D** and **E**, lane 3). If (**D**) ABI1 or (**E**) PP2CA together with staurosporine are added after the initial 10 min CPK6 mediated phosphorylation period, the SLAC1-NT phosphorylation signal rapidly decreases within 1 min (**D** and **E**, lanes 4–7). Staurosporine pre-exposure control inhibits SLAC1-NT phosphorylation by CPK6 (**D** and **E**, lane 8). (**F–H**) PP2Cs de-phosphorylate the SLAC1-NT which was phosphorylated by major SLAC1-activating kinases CPK23 and OST1. The SLAC1-NT is phosphorylated by CPK23 (**G**, lane 1) and OST1 (**H**, lane 1) which is inhibited when the PP2Cs ABI1 and PP2CA are added before starting the reactions (**G** and **H**, lanes 2–3). When adding staurosporine and elution buffer after the initial phosphorylation period and incubating for 10 min the signal does not change (**G** and **H**, lane 4). Addition of ABI1 or PP2CA after supplementing the reaction with staurosporine leads to rapid (10 min) dephosphorylation of the SLAC1-NT previously phosphorylated by the OST1 and CPK23 protein kinases (**G** and **H**, lanes 5–6).

The following figure supplement is available for figure 4:

**Figure supplement 1**. When previously phosphorylated by CPK21, the SLAC1-NT is de-phosphorylated by the PP2Cs ABI1 and PP2CA.

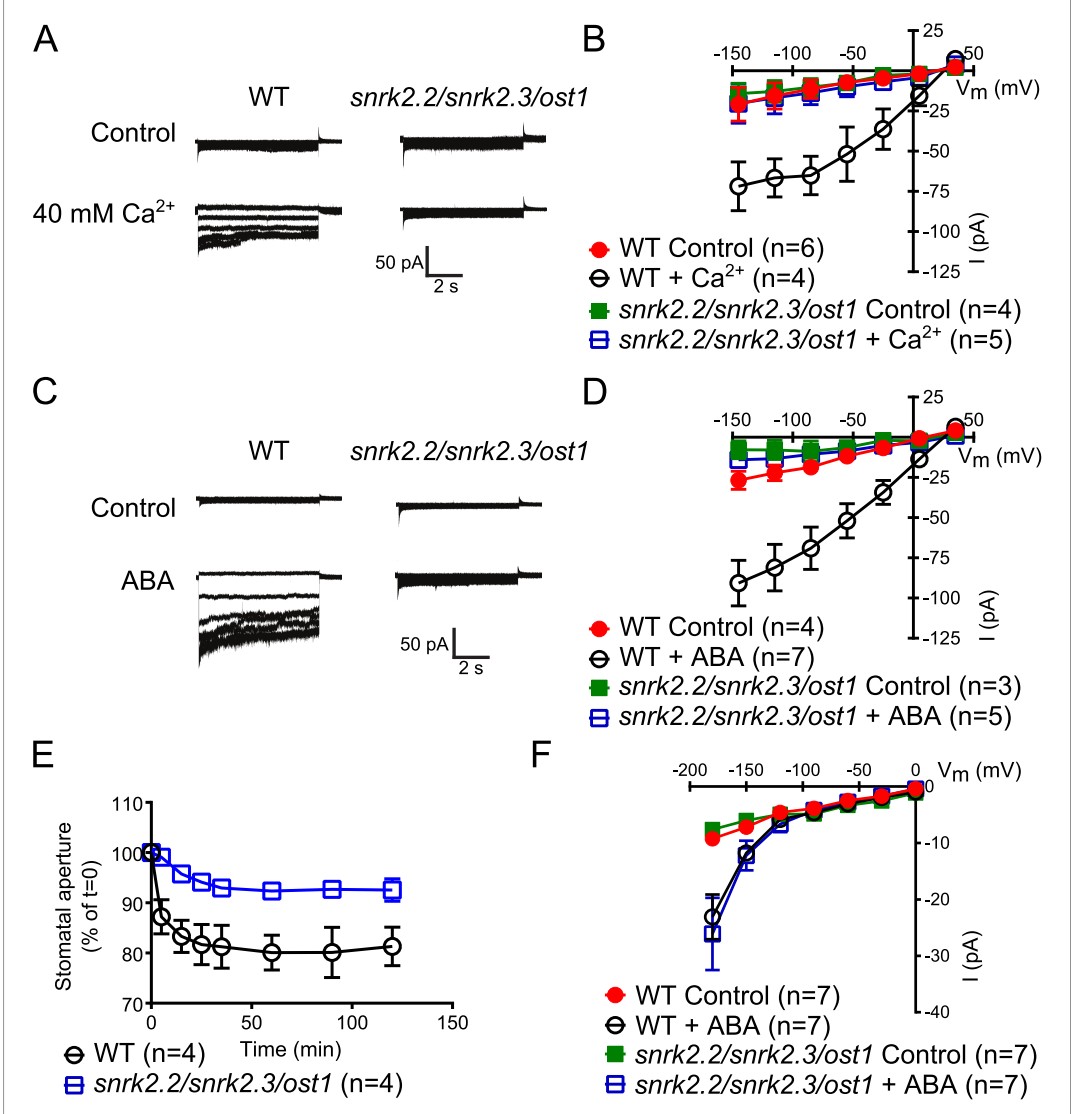

**Figure 5**. Both, ABA- and high external $Ca^{2+}$-activation of S-type anion currents at elevated $[Ca^{2+}]_{cyt}$ and imposed $Ca^{2+}$-oscillation-triggered stomatal closure are impaired in *snrk2.2/2.3/ost1* triple mutant guard cells while the ABA-activation of $I_{Ca}$ currents is intact. (**A–D**) Whole-cell patch-clamp experiments reveal that $[Ca^{2+}]_{cyt}$-activation of S-type anion currents is disrupted in *snrk2.2/2.3/ost1* triple mutant guard cells even if pre-incubated with high external $Ca^{2+}$ shock (**A** and **B**) or ABA (**C** and **D**). Note that pre-incubation with high external $Ca^{2+}$ shock by passes early ABA signaling (*Allen et al., 1999a*; *Allen et al., 2002*). Typical current responses (**A** and **C**), average steady-state current–voltage relationships ±SEM, and the number of measured cells are presented (**B** and **D**). In (**B**) data for *snrk2.2/2.3/ost1* triple mutants with and without ABA overlap with WT controls. (**E**) Imposed $Ca^{2+}$ oscillation-induced stomatal closure is impaired in $Ca^{2+}$-independent protein kinase *snrk2.2/2.3/ost1* triple mutant leaves, providing further evidence for an interdependence of these responses. Four 5-min extracellular $Ca^{2+}$-pulses were applied in 10-min intervals from time = 0 to 35 min. Average individually tracked stomatal apertures were normalized to the stomatal apertures at time zero. The averages of the normalized apertures ±SEM and the number of independent genotype-blind experiments (n = 4) are shown (>40 total stomata per group). Average stomatal apertures at time zero were 4.61 ± 0.44 μm in WT (n = 4) and 5.51 ± 0.87 μm in the *snrk2.2/2.3/ost1* triple mutant (n = 4). (**F**) Patch clamp experiments reveal that ABA activation of $I_{Ca}$ currents is not impaired in *snrk2.2/2.3/ost1* triple mutant guard cells. Average steady-state current–voltage relationships ±SEM, and the number of measured cells are presented in (**F**). Representative whole cell current traces for (**F**) are presented in *Figure 5—figure supplement 1*. Several error bars are not visible, as these were smaller than the illustrated symbols.

*Figure 5. continued on next page*

*Figure 5. Continued*

The following figure supplement is available for figure 5:

**Figure supplement 1**. *snrk2.2/2.3/ost1* triple mutant guard cells show intact ABA activation of $Ca^{2+}$-permeable ICa currents.

disruption of $Ca^{2+}$-independent signaling in *snrk2* triple mutants also impairs $Ca^{2+}$-dependent stomatal responses. Thus these findings investigating S-type anion channel regulation and stomatal movements both provide genetic evidence for an unexpected interdependence of the $Ca^{2+}$-dependent and -independent branches of the guard cell signaling network.

The $Ca^{2+}$-independent OST1 protein kinase affects $Ca^{2+}$ signaling in Landsberg *erecta* guard cells via regulation of plasma membrane-localized $Ca^{2+}$-permeable channels ($I_{Ca}$) (*Acharya et al., 2013*). To test whether the functional linkage of the $Ca^{2+}$-dependent and $Ca^{2+}$-independent branch is due to the regulation of the $I_{Ca}$ channels by SnRK2 protein kinases in the Columbia ecotype, we performed patch clamp analyses measuring plasma membrane $I_{Ca}$ channel currents in *snrk2.2/snrk2.3/ost1* triple mutant guard cells. However, ABA activation of $I_{Ca}$ channels remained intact in *snrk2.2/snrk2.3/ost1* triple mutant guard cells (*Figure 5F* and *Figure 5—figure supplement 1*). In positive control experiments, ABA receptor *pyr1/pyl1/2/4* quadruple mutant guard cells showed clear impairment of ABA activation of $I_{Ca}$ channels (Data not shown, n = 5; control vs ABA, p = 0.96; Student's *t*-test), consistent with previous findings (*Wang et al., 2013*).

## ABA-dependent stomatal responses are impaired in non-phosphorylatable SLAC1 serine 59 and serine 120 double mutant plants

In addition to possible direct cross-regulation of CPKs and SnRK2s, another non-mutually exclusive potential mechanism for the requirement of both SnRK and CPK kinases for ABA activation of S-type anion channels could be that SLAC1 serves as coincidence detector through differential phosphorylation by protein kinases of the $Ca^{2+}$-dependent and -independent branches. The amino acid residue serine 120 of SLAC1 has been shown to be required for OST1, but not for CPK23 activation of SLAC1 in *Xenopus* oocytes (*Geiger et al., 2009, 2010*). A different site, serine 59, has been shown to be required for SLAC1 activation by CPK6 (*Brandt et al., 2012*). Thus we investigated whether several CPKs can activate the SLAC1 S120A mutant in oocytes and whether the SLAC1 S59A mutant is activated by OST1 and other CPKs in oocytes. CPK5, CPK6, and CPK23 activation of SLAC1 S120A was similar to WT SLAC1 activation (*Figure 6A–C* and *Figure 6—figure supplement 1A,B*). In contrast, SLAC1 S59A activation by these CPKs was strongly impaired (*Figure 6A–C* and *Figure 6—figure supplement 1A,B*). Interestingly however, OST1 was able to activate SLAC1 S59A (*Figure 6D–F*), which was confirmed in multiple independent experimental sets under the imposed conditions. These results suggest that S59 is required for strong activation by protein kinases of the $Ca^{2+}$-dependent CPK branch, while S120 represents a crucial amino acid for strong activation by the $Ca^{2+}$-independent branch of the ABA signaling core. To avoid spurious phosphorylation by high protein kinase concentrations in oocytes, effects of co-expression of CPK6 and OST1 at low levels that do not fully activate SLAC1 were investigated. These experiments show a clear enhanced SLAC1 activation in oocytes when both kinases are co-expressed (*Figure 6—figure supplement 2A–D*). This enhancement of SLAC1 activation by OST1 became less clear when an inactive OST1 protein kinase (OST1 D140A) was analyzed (*Figure 6—figure supplement 2E*).

To more directly investigate S-type anion channel regulation *in planta*, we established *slac1-1* plant lines which express SLAC1 WT, S59A, S120A, and S59A/S120A fused to mVenus under the native SLAC1 promoter and carried out patch clamp analyses. Expression of wildtype SLAC1-mVenus in *slac1-1* guard cells resulted in recovery of S-type anion channels (*Figure 6G* and *Figure 6—figure supplement 3A*). Unexpectedly, expression of the single site SLAC1 mutants, SLAC1 S59A or SLAC1 S120A in *slac1-1* guard cells restored ABA regulation of S-type anion currents (*Figure 6G* and *Figure 6—figure supplement 3A*). However, expression of the double phosphorylation site SLAC1 mutant, SLAC1 S59A/S120A did not restore ABA activation of S-type anion channels (*Figure 6G* and *Figure 6—figure supplement 3A*). Furthermore, ABA-induced stomatal closing responses in these complementation lines confirmed the need to mutate both the S59 and S120 sites to alanine to

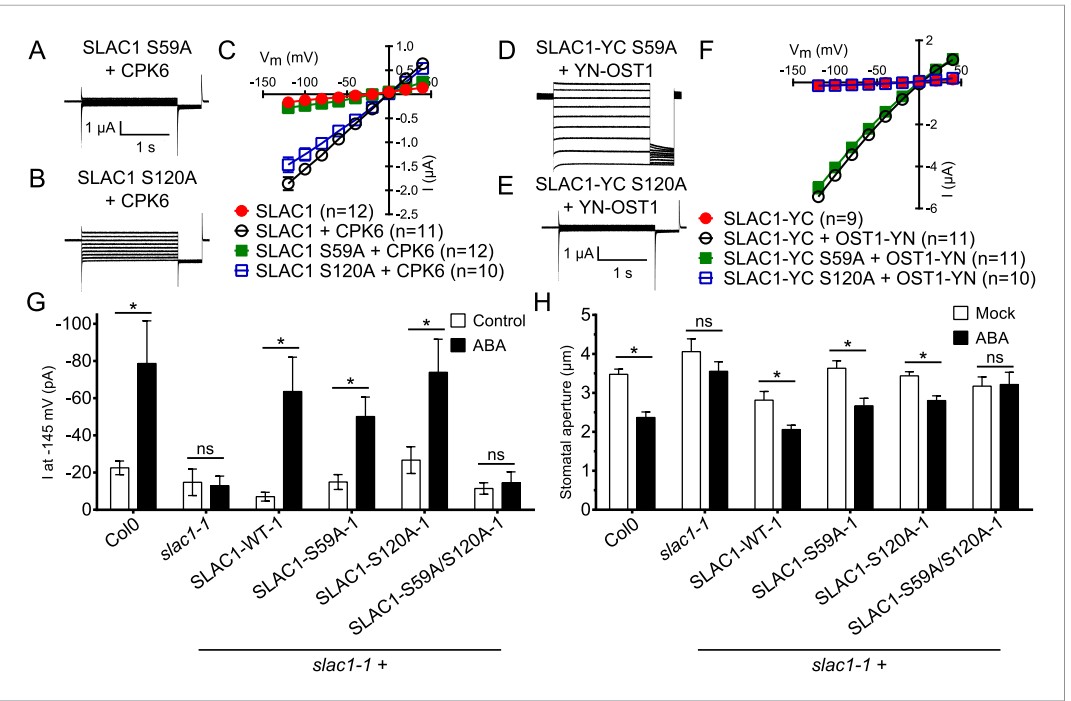

**Figure 6**. Ca²⁺-dependent protein kinase and OST1 protein kinase activation of SLAC1 in oocytes requires serine 59 or serine 120, respectively while *in planta* ABA-dependent S-type anion current activation and stomatal closing are only impaired in SLAC1 S59A/S120A double amino acid mutants. (**A–C**) SLAC1 activation by CPK6 in Xenopus oocytes was abolished when serine 59 is mutated to alanine (S59A) (**A** and **C**) (*Brandt et al., 2012*) but was comparable to wild type SLAC1 activation for the SLAC1 S120A mutated version (**B** and **C**). (**D–F**) OST1 activation of SLAC1 was abolished in the SLAC1 S120A mutant (**E** and **F**) (*Geiger et al., 2009*), while OST1 robustly activated SLAC1 S59A (**D** and **F**). (**G**) In whole-cell patch-clamp experiments, *slac1-1* guard cells show impaired ABA-activation of S-type anion currents. Expression of SLAC1 WT, S59A, and S120A in *slac1-1* plants restores ABA activation of S-type anion currents in guard cells, but expression of SLAC1 S59A/S120A does not. (**H**) The ABA-insensitive phenotype of *slac1-1* stomata was recovered by expression of SLAC1 WT, S59A, and S120A, but not by expression of S59A/S120A. Note that SLAC1 WT, S59A, S120A, and S59A/S120A are expressed as C-terminal mVenus fusion proteins under native SLAC1 promoter (see 'Materials and methods'). Representative current traces are depicted in (**A**, **B**, **D** and **E**) and average current voltage relationships are shown (**C** and **F**; ±SEM). Average steady-state current responses ±SEM at −145 mV are plotted in (**G**) and average stomatal apertures ±SEM in (**H**). * indicates p < 0.05; unpaired Student's *t*-test. Exact p-values and number of individual experiments for (**G** and **H**) can be found in *Figure 6—figure supplement 4*. Note that WT (Col0) and *slac1-1* control measurements shown in (**G** and **H**) are the same control data as those shown in *Figure 6—figure supplement 3A,B* as all lines were investigated under the same conditions. Several error bars are not visible, as these were smaller than the illustrated symbols.

The following source data and figure supplements are available for figure 6:

**Source data 1**. Statistical data and number of repeats (n) for the (Table 1) patch clamp measurements shown in *Figure 6G* and *Figure 6—figure supplement 3A* and (Table 2) for measurements of stomatal apertures presented in *Figure 6H* and *Figure 6—figure supplement 3B* (n = 3 experiments and >45 total stomata per group).

**Figure supplement 1**. SLAC1 serine 59 but not serine 120 is required for CPK5 or CPK23 activation in Xenopus oocytes.

**Figure supplement 2**. SLAC1 exhibits enhanced activity by co-expression of CPK6 and OST1 in Xenopus oocytes.

**Figure supplement 3**. ABA-induced S-type anion currents and stomatal closure responses are impaired when both SLAC1 S59 and S120 are substituted with alanine in independent double amino acid mutant line.

**Figure supplement 4**. Analysis of expression and subcellular localization of SLAC1-WT, SLAC1S59A, S120A, and S59A/S120A in *slac1-1* complementation lines.

*Figure 6. continued on next page*

*Figure 6. Continued*

**Figure supplement 5**. BiFC fluorescence intensities are altered for CPK6 and ABI1 co-expression with SLAC1-WT, SLAC1S59A, S120A, and S59A/S120A.

significantly impair ABA-induced stomatal closing *in planta* (*Figure 6H* and *Figure 6—figure supplement 3B*). The above described patch clamp and stomatal movement experiments were conducted with two independent complementation lines (*Figure 6G,H* and *Figure 6—figure supplement 3A,B*). To ensure that the impaired ABA-activation of S-type anion currents and stomatal closure in the SLAC1 S59A/S120A mutant was not due to non-expressed protein we investigated the mVenus-derived fluorescence in all complementation lines. All SLAC1 complementation lines expressed SLAC1-mVenus driven by the native *SLAC1* promoter to a similar degree (*Figure 6—figure supplement 4*).

We examined putative roles of the two phosphorylation sites in SLAC1 for interaction of SLAC1 with CPK6 and ABI1 by BiFC analysis. Reconstituted YFP fluorescence intensity of CPK6-YN co-expressed with YC-SLAC1-S59A, YC-SLAC1-S120A, or YC-SLAC1-S59A/S120A was significantly lower than that of CPK6-YN co-expressed with YC-SLAC1-WT (*Figure 6—figure supplement 5*). YN-ABI1 co-expression with the YC-SLAC1-S59A mutant did not significantly change the YFP fluorescence intensity while YN-ABI1 co-expression with YC-SLAC1-S120A or YC-SLAC1-S59A/S120A resulted in lower YFP fluorescence intensity when compared to YN-ABI1 co-expression with YC-SLAC1-WT ($p < 0.005$; unpaired t-test; *Figure 6—figure supplement 5*). These results point to the need for future research to determine whether these phosphorylation sites in SLAC1 might contribute to promotion of CPK6 kinase and ABI1 phosphatase interaction strength with the SLAC1 channel (*Figure 6—figure supplement 5*).

## Discussion

Dissection of $Ca^{2+}$ signaling specificity mechanisms can be advanced through characterization of the combined cellular, genetic, and biochemical mechanisms in a single cell type. Biochemical and cellular mechanisms that function in $Ca^{2+}$ specificity, notably those mediated by the mammalian $Ca^{2+}$/calmodulin-dependent kinase II, have been characterized (*De Koninck and Schulman, 1998*; *Bradshaw et al., 2003*; *Rellos et al., 2010*; *Chao et al., 2011*). Genome analyses in plants have revealed the existence of more than 200 genes encoding for proteins containing $Ca^{2+}$-binding EF-hands in the *Arabidopsis* genome alone (*Day et al., 2002*) with overlapping expression of many genes in the same cell type, including guard cells (*Harmon et al., 2000*; *McCormack et al., 2005*; *Schmid et al., 2005*; *Winter et al., 2007*). This plethora of $Ca^{2+}$ signaling proteins and the many responses in plants mediated by $Ca^{2+}$ (*Dodd et al., 2010*) calls for robust mechanisms mediating specificity in $Ca^{2+}$ signaling.

$Ca^{2+}$ is a major hub within the signaling network of plant guard cells (*MacRobbie, 2000*; *Hetherington, 2001*; *Hetherington and Woodward, 2003*), but the biochemical mechanisms mediating $Ca^{2+}$ specificity have remained unknown. In guard cells, stomatal closing stimuli, including ABA and $CO_2$, enhance (prime) $[Ca^{2+}]_{cyt}$-sensitivity, as also shown in intact *Arabidopsis* and *V. faba* guard cells (*Young et al., 2006*; *Munemasa et al., 2007*; *Siegel et al., 2009*; *Chen et al., 2010*; *Xue et al., 2011*). Calcium sensitivity priming could provide a key mechanism contributing to specificity in $Ca^{2+}$ signaling, as this response switches between a state of reduced $Ca^{2+}$ sensitivity to a $Ca^{2+}$-responsive 'primed' state, thus tightly controlling $Ca^{2+}$ responsiveness (*Allen et al., 2002*; *Young et al., 2006*; *Munemasa et al., 2007*; *Siegel et al., 2009*; *Chen et al., 2010*; *Xue et al., 2011*). However, the genetic and biochemical mechanisms mediating $Ca^{2+}$ sensitivity priming have remained unknown.

Here we report genetic, biochemical and signaling network mechanisms that underpin this cellular response. In the absence of ABA, $Ca^{2+}$ responsiveness is inhibited by PP2Cs, thereby preventing responses to unrelated stomatal opening-mediating stimuli (*Irving et al., 1992*; *Shimazaki et al., 1992*; *Curvetto et al., 1994*; *Shimazaki et al., 1997*; *Cousson and Vavasseur, 1998*; *Young et al., 2006*) and also spontaneous $Ca^{2+}$ elevations (*Young et al., 2006*; *Siegel et al., 2009*). As PP2Cs inhibit OST1 and also down-regulate SLAC1 directly, this network not only enables stimulus specific activation of SLAC1, but also provides a tight off switch via PP2C-catalyzed dephosphorylation of

SLAC1 (*Figure 7*). This mechanism could also prevent SLAC1 activation by CPK23 which exhibits a moderate $Ca^{2+}$ sensitivity (*Geiger et al., 2010*). Moreover, as PP2Cs control $Ca^{2+}$ signaling specificity downstream of the CPK $Ca^{2+}$ sensors (*Figure 7*), the same CPK isoforms remain capable of fulfilling other signaling roles, consistent with several studies (*Boudsocq et al., 2010*; *Munemasa et al., 2011*; *Dubiella et al., 2013*; *Gao et al., 2013*). Similarly, the same MAP kinase genes have been shown to function in multiple plant signaling pathways and unknown mechanisms mediating specificity are required (*Rodriguez et al., 2010*; *Xu and Zhang, 2015*). It was reported that ABI1 is not able to remove phosphate groups from SLAC1 after OST1 phosphorylation (*Geiger et al., 2009*; *Scherzer et al., 2012*). In contrast, the present study and other recent research shows a clear dephosphorylation of SLAC1 by PP2Cs (*Brandt et al., 2012*; *Maierhofer et al., 2014*). Here we demonstrate that ABI1 and PP2CA not only dephosphorylate the SLAC1 N-terminus, but these PP2Cs are able to very rapidly remove the OST1- and CPK-mediated phosphorylation of SLAC1 (*Figure 4C–H* and *Figure 4—figure supplement 1*).

The $Ca^{2+}$-dependent and $Ca^{2+}$-independent ABA-signaling branches are presently considered to function independent of one another (e.g., *Li et al., 2006*; *Kim et al., 2010*; *Roelfsema et al., 2012*). However, this model has not yet been investigated using higher order genetic mutants. In the present study we unexpectedly have found that *snrk2.2/snrk2.3/ost1* triple mutant plants in the $Ca^{2+}$-independent ABA signal transduction pathway, also abrogate the 'by pass' (*Allen et al., 1999a*) $Ca^{2+}$-induced $[Ca^{2+}]_{cyt}$ activation of S-type anion channels and $Ca^{2+}$ oscillation-induced stomatal closing *in planta* (*Figure 5A,B,E*). These data show an unexpected dependence of $Ca^{2+}$-dependent stomatal closing on the $Ca^{2+}$-independent SnRK2 protein kinase signaling branch. Moreover, we have identified the *cpk5/6/11/23* quadruple mutations to impair ABA activation of S-type anion channels and stomatal closure (*Figure 1*). Notably, this impairment occurs despite an intact SnRK2 signaling branch. Note that a weakened ABA-induced stomatal closing response in *cpk* mutant plants, as found here when applying 10 µM ABA (*Figure 1F*), is likely the result of parallel ABA activation of R-type anion channels (*Meyer et al., 2010*; *Sasaki et al., 2010*; *Imes et al., 2013*) and a possible less-stringent CPK regulation of the SnRK2 signaling branch. In *cpk5/6/11/23* quadruple mutant plants signal

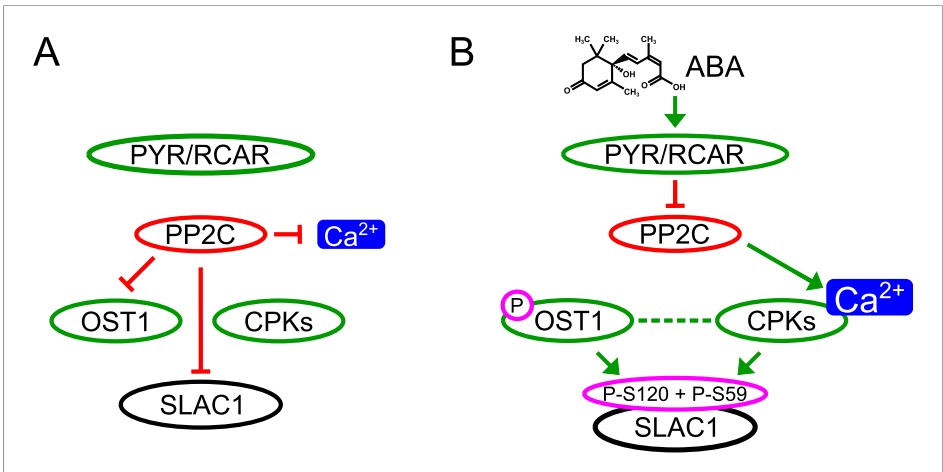

**Figure 7**. Simplified schematic model for $Ca^{2+}$-specificity mechanism within ABA-dependent SLAC1 activation in guard cells. (**A**) Without ABA, $Ca^{2+}$ elevations that can also function in stomatal opening responses (*Irving et al., 1992*; *Shimazaki et al., 1992*; *Curvetto et al., 1994*; *Shimazaki et al., 1997*; *Cousson and Vavasseur, 1998*; *Young et al., 2006*) and spontaneous or un-specifically induced $Ca^{2+}$ transients (*Allen et al., 1999b*; *Klüsener et al., 2002*; *Young et al., 2006*; *Yang et al., 2008*; *Siegel et al., 2009*) do not lead to S-type anion channel (SLAC1) activation as PP2C protein phosphatases directly negatively regulate SLAC1 activation. (**B**) In the presence of ABA this SLAC1 inhibition is released, OST1 and CPKs phosphorylate, and thereby activate the channel. ABA also causes $[Ca^{2+}]_{cyt}$ elevation via PP2C inhibition (*Allen et al., 1999a*; *Murata et al., 2001*). Data indicate cross-talk between $Ca^{2+}$-dependent and -independent ABA-activation of SLAC1 which may be mediated through a combination of protein kinase cross regulation and additive activation via differential affinities for SLAC1 phosphorylation sites by OST1 and CPKs.

transduction via the $Ca^{2+}$-independent SnRK2 pathway appears to partially prevail at higher ABA concentrations. Together these data indicate an unexpected dependence of the $Ca^{2+}$-dependent signal transduction pathway on the $Ca^{2+}$-independent SnRK2 protein kinase-mediated pathway (*Figure 5*). Furthermore, the present results together indicate that the output of the $Ca^{2+}$-dependent signaling pathway may affect the output of the SnRK2 signaling branch.

The presented combined genetic, cell signaling and physiological response analyses provide strong evidence for a concomitant requirement of both the $Ca^{2+}$-dependent and $Ca^{2+}$-independent branches to trigger a robust (*Hetherington, 2001*) downstream stomatal closing response (*Figure 7*). One model for cross talk of SnRK2-induced signaling with $Ca^{2+}$ signaling could be that OST1 causes the activation of the $Ca^{2+}$-permeable plasma membrane $I_{Ca}$ channels (*Hamilton et al., 2000*; *Pei et al., 2000*). However, our data clearly show that triple knock out of the $Ca^{2+}$-independent SnRK2 kinases, OST1, SnRK2.2, and SnRK2.3 in the Columbia accession, does not impair ABA activation of $I_{Ca}$ channels (*Figure 5F* and *Figure 5—figure supplement 1*). Interestingly however, *cpk* mutants show impairment in ABA activation of $I_{Ca}$ channels in guard cells (*Mori et al., 2006*).

The present study suggests that the integration of signals via differential phosphorylation of SLAC1 by the kinases of the $Ca^{2+}$-dependent and $Ca^{2+}$-independent branches could contribute to the interdependence of both signaling branches. In *Xenopus* oocytes, SLAC1 S59 is required for the activation by CPKs while SLAC1 S120 is required for the activation by the $Ca^{2+}$-independent kinase OST1 in oocytes (*Figure 6A–F* and *Figure 6—figure supplement 1*). Additionally, SLAC1 activation is enhanced by co-expression of (non-split YFP moieties) non-saturating OST1 and CPK6 activities (*Figure 6—figure supplement 2*). However, *in planta* analyses of *slac1-1* plants expressing single SLAC1 S59A or SLAC1 S120A mutants under the control of the *SLAC1* promoter unexpectedly display intact ABA-responses indicating that the phosphorylation of either amino acid residue, together with phosphorylation of other amino acids, is sufficient for ABA-induced stomatal closing in intact stomata and ABA activation of S-type anion channels (*Figure 6G,H* and *Figure 6—figure supplement 3*). Furthermore, simultaneous mutation of both residues in SLAC1 (S59A and S120A) caused a strong impairment in ABA activation of S-type anion channels and stomatal closing *in planta*, illustrating the combined key functions of these residues in the intact guard cell system.

It should be noted that although SLAC1 S120, but not S59, is crucial for the activation by OST1 in *Xenopus* oocytes (*Figure 6E,F*) (*Geiger et al., 2009*), phosphorylation of SLAC1 S59 by OST1 is also found in vitro (*Vahisalu et al., 2010*). In addition, although the S120A mutation does not affect CPK6 activation of SLAC1 in *Xenopus* oocyte system (*Figure 6B,C*), our LC-MS/MS analyses reveal that the S120 can be also phosphorylated by CPK6 in vitro (data not shown). Combined with these in vitro data, our present *in planta* findings suggest that the SnRK2 and CPK protein kinases may have distinct affinities for the S59 and S120 phospho-sites of SLAC1, which could contribute to the interdependence of the $Ca^{2+}$-dependent and -independent branches of the ABA signaling network. In addition, crosstalk regulation mechanisms of these protein kinase responses may exist *in planta* and will require further investigation (*Figure 7*).

Note that, similar to the *slac1-1* mutation, mutation of SLAC1 S120 to phenylalanine (*slac1-7*) can impair ozone-induced stomatal closing (*Vahisalu et al., 2010*). It is plausible that a phenylalanine residue at this position causes more significant structural changes that impair SLAC1 function compared to alanine. When both S59 and S120 are mutated to alanine simultaneously however, ABA-triggered S-type anion current activation and stomatal closure were abrogated, highlighting the importance of these two residues for ABA-signaling *in planta*. The results gained *in planta* also highlight that data gained in oocytes, though helpful, are simplified and, not surprisingly, do not necessarily represent the situation in the complex plant system. Over-expression of the components, including activating protein kinases, to a high abundance in oocytes is well-suited to test several possible mechanisms in ion channel regulation, and can guide follow up investigation in the native environment in plant cells.

## Conclusions

In summary, the present study reveals a first genetic mechanism that mediates $Ca^{2+}$ sensitivity priming. $Ca^{2+}$ sensitivity is demonstrated here to be constitutively primed in *pp2c* quadruple mutant guard cells, showing that PP2Cs ensure $Ca^{2+}$ signaling specificity. Interestingly, PP2Cs do not directly down-regulate CPK activity, in contrast to direct PP2C down-regulation of the SnRK2 protein kinases. Rather PP2Cs very rapidly down-regulate signaling targets downstream of CPKs, which could enable

the same CPKs to function in more than one pathway. We have further identified a *cpk* quadruple mutant here that for the first time strongly abrogates ABA activation of S-type anion channels. This abrogation occurs despite an intact $Ca^{2+}$-independent SnRK2 signaling branch. Furthermore, disruption of the $Ca^{2+}$-independent signaling branch in *snrk2* protein kinase triple mutant plants abrogates $Ca^{2+}$ signaling. Thus, unexpectedly genetic analyses reveal a dependence of the $Ca^{2+}$-sensitive ABA signaling branch on the $Ca^{2+}$-insensitive branch *in planta*. The control of ABA-triggered stomatal closure by parallel interdependent $Ca^{2+}$-dependent and–independent mechanisms could contribute to the robustness of the stomatal ABA signaling network (*Hetherington, 2001*). Unexpectedly, *in planta* studies show that the S59 and S120 phosphorylation sites in SLAC1 are together required for intact ABA-induced stomatal closing in vivo. The $Ca^{2+}$ sensitivity priming mechanism described here could represent a more general principle present in plants contributing to $Ca^{2+}$ specificity within cellular signal transduction pathways, while also maintaining the availability of $Ca^{2+}$ sensors for distinct $Ca^{2+}$-dependent signaling outputs.

## Materials and methods

### Mutant plant lines

All *A. thaliana* plants used in this study are in the Col0 ecotype. *cpk5/6/11/23* quadruple T-DNA insertion mutant plants were established by crossing *cpk5/6/11* (sail_657C06/salk_025460/salk_054495) kindly provided by Dr Jen Sheen (Harvard Medical School) (*Boudsocq et al., 2010*) with *cpk23-1* (salk_007958) obtained from ABRC (*Ma and Wu, 2007*; *Geiger et al., 2010*). Dr Ping He (Texas A&M University) shared *cpk1/2/5/6* (salk_096452/salk_059237/sail_657C06/salk_025460) mutant seeds (*Gao et al., 2013*). The PP2C quadruple knock-out plants (*abi1-2/abi2-2/hab1-1/pp2ca-1*; salk_072009/salk_015166/salk_002104/salk_028132) and *snrk2.2/2.3/ost1* (GABI-Kat_807G04/salk_107315/salk_008068) were kindly provided by Dr Pedro L Rodriguez (University of Valencia) (*Antoni et al., 2013*). A second independent *snrk2.2/2.3/ost1* (GABI-Kat_807G04/salk_107315/salk_008068) line was established by crossing *snrk2.2/2.3* supplied by Dr Jian-Kang Zhu (Shanghai Center for Plant Stress Biology) with *ost1-3*. To establish SLAC1 complementation lines a 4.4 kb fragment including 1.63 kb of the 5′-UTR, the genomic *SLAC1* gene region and 0.9 kb of the 3′-UTR (*Negi et al., 2008*) was amplified using the PfuX7 polymerase (*Norholm, 2010*). The fragment was cloned into a modified pGreenII (*Hellens et al., 2000*) vector lacking a promoter and being compatible with USER-cloning. Employing USER cloning (*Nour-Eldin et al., 2006*; *Bitinaite et al., 2007*; *Geu-Flores et al., 2007*) the point mutations were introduced and *SLAC1* was fused with mVenus (C-terminally) (*Nagai et al., 2004*). These pGreenII constructs were transformed into *Agrobacterium tumefaciens* GV3101(pMP90) RG (*Koncz and Schell, 1986*). *slac1-1* mutant plants were then transformed by the floral dipping method (*Clough and Bent, 1998*) and propagated until the T-DNA insertion was confirmed to be homozygous.

### Patch clamp analyses

*Arabidopsis* plants were grown on soil in the growth chamber at 21°C under a 16-hr-light/8-hr-dark photoperiod with a photon flux density of 80 μmol/($m^2 \times s$). The plants were watered from bottom trays with deionized water once or twice per week and sprayed with deionized water every day. The growth chamber humidity was 50–70%.

*Arabidopsis* guard cell protoplasts were isolated enzymatically as previously described (*Pei et al., 1997*). One or two rosette leaves of 4- to 5-week-old plants were blended in a blender with deionized water at room temperature (RT) for approximately 30 s. For isolation of guard cell protoplasts from *snrk2.2/snrk2.3/ost1* triple mutants, four or five rosette leaves were used. Epidermal tissues were collected using a 100-μm nylon mesh and rinsed well with deionized water. The epidermal tissues were then incubated in 10 ml of enzyme solution containing 1% (wt/vol) Cellulase R-10 (Yakult, Japan), 0.5% (wt/vol) Macerozyme R-10 (Yakult, Japan), 0.1 mM KCl, 0.1 mM $CaCl_2$, 500 mM D-mannitol, 0.5% (wt/vol) BSA, 0.1% (wt/vol) kanamycin sulfate, and 10 mM ascorbic acid for 16 hr at 25°C on a circular shaker at 40 rpm. Guard cell protoplasts were then collected by filtering through a 20-μm nylon mesh. Subsequently, the protoplasts were washed twice with washing solution containing 0.1 mM KCl, 0.1 mM $CaCl_2$, and 500 mM D-sorbitol (pH 5.6 with KOH) by centrifugation for 10 min at 200×*g*. The guard cell protoplast suspension was kept on ice before use.

To investigate ABA activation of S-type anion channels, the guard cell protoplast suspension was pre-incubated with 10 μM (*Figure 1A,C*, *Figure 1—figure supplement 1C,D*, *Figure 6G*, and *Figure 6—figure supplement 3A*) or 50 μM (*Figure 2A–D* and *Figure 2—figure supplement 1A,B* as well as *Figure 5C,D*) ± ABA (Sigma, St. Louis, MO) for 30 min. S-type anion channel currents in guard cell protoplasts were recorded by the whole-cell patch-clamp technique as previously described (*Pei et al., 1997*; *Vahisalu et al., 2008*; *Siegel et al., 2009*). The pipette solution contained 150 mM CsCl, 2 mM $MgCl_2$, 5 mM Mg-ATP, 6.7 mM EGTA, and 10 mM Hepes-Tris (pH 7.1). To obtain a free $[Ca^{2+}]_{cyt}$ of 2 μM and 110 nM, 5.86 mM and 1.79 mM of $CaCl_2$ were added to the pipette solution, respectively. Osmolality of the pipette solution was adjusted to 500 mmol/l using D-sorbitol. The bath solution contained 30 mM CsCl, 2 mM $MgCl_2$, 1 mM $CaCl_2$, and 10 mM MES-Tris (pH 5.6). Osmolality of the bath solution was adjusted to 485 mmol/l using D-sorbitol. To investigate external $Ca^{2+}$ activation of S-type anion channels, guard cell protoplasts were pre-incubated with the bath solution containing 40 mM $CaCl_2$, instead of 1 mM $CaCl_2$ for 30 min. Whole-cell currents were recorded 3–5 min after achieving the whole-cell configuration. The seal resistance was no less than 10 GΩ. The voltage was decreased from +35 mV to −145 mV with 30 mV decrements and the holding potential was +30 mV.

To investigate ABA activation of $Ca^{2+}$-permeable $I_{Ca}$ channels, the pipette solution contained 10 mM $BaCl_2$, 4 mM EGTA, and 10 mM HEPES-Tris (pH 7.1). 5 mM NADPH was freshly added to the pipette solution before experiments. The bath solution contained 100 mM $BaCl_2$, and 10 mM MES-Tris (pH 5.6). 0.1 mM DTT was freshly added to the bath solution before experiments. Osmolarity was adjusted to 500 mmol/l for the pipette solution and 485 mmol/l for the bath solution with D-sorbitol. A ramp voltage protocol from +20 to −180 mV (holding potential, 0 mV; ramp speed, 200 mV/s) was used for $I_{Ca}$ recordings (*Pei et al., 2000*). The seal resistance was no less than 10 GΩ. Data were filtered at 3 kHz. Initial control whole-cell currents were recorded 10 times with a 1 min interval between each recording 1–3 min after achieving whole-cell configurations. The average current obtained from the 10 current traces per cell at 0, −30, −60, −90, −120, −150, and −180 mV was determined for IV curves. After control current recordings, ABA was added to the bath solution by perfusion, and guard cell protoplasts were incubated with ABA in the bath solution for 3 min. Then, ABA-activated $I_{Ca}$ currents were recorded 10 times for another 10 min and the average current obtained from the 10 traces was determined for IV curves.

## Stomatal aperture analyses

2-week-old plate-grown plants were transferred to soil and grown in >70% relative humidity under 16 hr light/8 hr dark. Rosette leaves from 4- to 5-week-old plants were detached and incubated in stomatal opening buffer (5 mM KCl, 50 μM $CaCl_2$, 10 mM MES and pH 5.6 with Tris base) for 2.5 hr, in 150–180 μmol/($m^2 \times$ s) light. Next, leaves were treated with either 5 μM ABA or 0.05% ethanol for an additional 1 hr incubation. After the incubation period, leaves were blended and fragments were collected with a 100 μm nylon mesh (*Figure 1E*, *Figure 6H* and *Figure 6—figure supplement 3B*) except for *Figure 1F*. In *Figure 1F*, epidermal peels were prepared using a perforated-tape epidermal detachment method (*Ibata et al., 2013*). Images of stomata from the abaxial side of the leaves were collected by microscopy. Stomatal aperture analyses were conducted as single-blind experiments in which the experimenter did not know the plant genotypes during measurements (*Figure 1E,F*) or as double-blind experiments in which the experimenter did not know both the ABA concentration and the plant genotypes (*Figure 6H* and *Figure 6—figure supplement 3B*).

## Imposed $Ca^{2+}$ pulse-regulated stomatal apertures of individually mapped stomata

Stomatal aperture analyses for imposed $Ca^{2+}$ pulses were performed as previously described (*Allen et al., 2001*; *Mori et al., 2006*; *Siegel et al., 2009*). Stomatal apertures of individually mapped stomata were measured at the indicated time points after the start of imposed $Ca^{2+}$ pulses. The lower epidermis of rosette leaves from 4- to 5-week-old plants was attached onto a coverslip using medical adhesive (Hollister). Then mesophyll layers of the leaf were carefully removed using a razor blade until only the epidermal layer remained. The lower epidermis was incubated in depolarizing buffer (50 mM KCl and 10 mM MES-Tris [pH 5.6]) for 3 hr under white light (150–180 μmol/($m^2 \times$ s)) to open stomata. Depolarizing buffer was changed to hyperpolarizing buffer (1 mM KCl, 1 mM $CaCl_2$, and 10 mM MES-Tris at pH 5.6). Four 5-min extracellular $Ca^{2+}$ pulses were applied in 5-min intervals in the first 35 min.

Stomatal aperture analyses were conducted as blind experiments in which the experimenter did not know the plant genotypes during measurements (*Figure 5E*).

## Recombinant protein isolation

Over-expression and purification of recombinant proteins were performed as described in *Brandt et al. (2012)* with minor adjustments: For the isolation of the PP2C proteins ABI1, ABI2, and PP2C additionally 5 mM MgCl$_2$ and 5% Glycerol were added to the buffer in which the bacterial pellet were re-suspended (buffer W in IBA manual). Also, all proteins except SLAC1-NT were eluted in elution buffer supplemented with 20% Glycerol instead of 10% and stored at −80°C instead of −20°C. To assess protein concentrations, several volumes of the eluates were loaded on a gel together with several defined bovine serum albumin (BSA) protein amounts. After separating the proteins by SDS-PAGE (*Laemmli, 1970*), the proteins were stained with coomassie brilliant blue R-250, dried between two sheets of cellophane, and then scanned. BSA and recombinant protein band intensities were measured using Fiji (*Schindelin et al., 2012*). After subtracting the background signal, BSA band signal intensities were used to plot a standard curve. Concentrations of isolated recombinant proteins were then calculated based on the equation resulting from the linear regression of the BSA standard curve.

## Whole plant protein extraction

Seeds were sterilized by incubation in sterilization medium (70% ethanol and 0.04% (wt/vol) SDS) for 15 min followed by three washes in 100% ethanol. After drying, the seeds for all genotypes were plated on one plate with ½ Murashige and Skoog Basal Medium (MS; Sigma–Aldrich, St. Louis, MO) and 0.8% phyto-agar. The plate was then stored at 4°C for >3 days and subsequently transferred to a growth cabinet (16/8 light/dark and 22°C). After a growth phase of 10–14 days >10 seedlings per genotype were floated on liquid ½ MS and equilibrated for 60–90 min in the growth cabinet. Either ± ABA (Sigma) to a final concentration of 50 µM (indicated by + in the figure) or the same volume of solvent control (ethanol; indicated by—in the figure) was added to the floating seedlings. After 30 min the seedlings were removed from the ½ MS and flash frozen in liquid nitrogen. Plant tissue was disrupted by shaking the frozen seedlings together with steel balls in a shaker (Retsch) for three times 30 s at 30 Hz in pre-cooled mountings. Subsequently, extraction buffer: 100 mM HEPES-NaOH pH 7.5, 5 mM EDTA, 5 mM EGTA, 0.5% (vol/vol) Triton X-100, 150 mM NaCl, 0.5 mM DTT, 10 mM NaF, 0.5% (vol/vol) protease inhibitor (Sigma–Aldrich), 0.5% (vol/vol) phosphatase inhibitor 2 (Sigma–Aldrich), 0.5% (vol/vol) phosphatase inhibitor 3 (Sigma–Aldrich), 5 mM Na$_3$VO$_4$, and 5 mM β-Glycerophosphate disodium salt hydrate was added. The samples were then treated in a sonication water bath (Fisher Scientific) with ice added to the water for 30 s. Cell debris was removed via centrifugation at 20,000×$g$ and 4°C for 40 min. Protein concentrations of the supernatants were measured using the BCA Protein Assay Kit (Pierce). 20 µg of total protein for each genotype and treatment were subjected to SDS-PAGE (*Laemmli, 1970*) under denaturing conditions (see in-gel kinase assay).

## In vitro protein kinase activity analyses

The reaction buffer consisted of 100 mM HEPES-NaOH pH 7.5, 10 mM MgCl$_2$, 2 mM DTT, 1 mM EGTA, and CaCl$_2$ was added to get a final concentration of 2.5 µM free Ca$^{2+}$ for all assays except the assay depicted in *Figure 3—figure supplement 4* for which free Ca$^{2+}$ was adjusted to 5 µM (calculated with http://www.stanford.edu/~cpatton/webmaxc/webmaxcE.htm). Note that the pH of the reaction buffer dropped to pH 7.3 after adding all components and free Ca$^{2+}$ calculations were performed accordingly. The flow charts in the respective figures indicate the components which were added subsequently in sequence (from top to bottom) and the respective incubation times. For the reactions shown in *Figure 3—figure supplement 4* 0.5 µg of CPK6 and 1 µg of the PP2Cs ABI1, ABI2, and PP2CA were used. The addition of EGTA for reactions shown in *Figure 3—figure supplement 4* lanes 2–4 resulted in a free Ca$^{2+}$ concentration <10 nM (calculated with http://www.stanford.edu/~cpatton/webmaxc/webmaxcE.htm). For the experiments shown in *Figure 4D–H* and *Figure 4—figure supplement 1*, SLAC1-NT (1.5 µg) was mixed together with 200 nM of the protein kinases CPK6, CPK23, OST1, and CPK21 in reaction buffer. Staurosporine was added to a final concentration of 100 µM and the final concentration of the PP2Cs ABI1 and PP2CA was 600 nM. To start all in vitro kinase reactions, 5 µCi of [γ-$^{32}$P]-ATP (Perkin–Elmer) was added and the reactions were incubated at RT for 10 min. The final volumes were 20 µl and the reactions were stopped by the

addition of 4 µl of 6× loading dye with subsequent incubation at 95°C for 5 min. The proteins were then separated by SDS polyacrylamide gel electrophoresis (SDS-PAGE, *Laemmli, 1970*) in 4–20% acryl amide gradient gels (Biorad). After, the proteins were stained with coomassie brilliant blue R-250 (Sigma). To visualize incorporated $^{32}$P-derived radioactive signals, gels were exposed to a storage phosphor screen (Molecular Dynamics; *Figure 4D–H* and *Figure 4—figure supplement 1*) or HyBlot CL autoradiography films (Denville Scientific; *Figure 3—figure supplement 4*). The phosphor storage screen was read out using a Typhoon scanner (Amersham Bioscience).

To compare CPK6 activities by measuring ATP consumption (*Figure 3—figure supplement 5*) with or without the PP2Cs ABI1 and PP2CA, 0.5 µM of the protein kinase was incubated at RT for 7.5 min either alone or with 1 µM of PP2C protein in the above mentioned reaction buffer supplemented with 10 µM ATP and ~150 µM Histone III-S (Sigma). The reactions were stopped by the addition of staurosporine. Residual ATP levels were quantified using the Kinase-Glo kit (Promega) according to the manufacturer's instructions resulting in luminescence signals measured in a plate reader (Berthold Mithras LB 940) (*Latz et al., 2012*). ATP consumption was calculated by first assessing the maximum range ($\Delta R_{max}$) of luminescence by subtracting the signal intensity of the background (no ATP added; $R_B$) from the maximum signal (no kinase added; $R_{max}$). To calculate the ATP consumption, signal intensities derived from the residual ATP in the reactions ($R_x$) were subtracted from the maximum signal ($R_{max}$) and then related to the maximum range ($\Delta R_{max}$) and plotted in per cent ($[(R_{max} - R_x)/\Delta R_{max}] \times 100$).

## In-gel kinase assays

For in-gel protein kinase assays using recombinant proteins shown in *Figure 3C,D*, 500 ng of OST1 and CPK6 kinase and the PP2Cs ABI1 and PP2CA in a 1:3 molar ration were mixed in reaction buffer with 2.5 µM free Ca$^{2+}$ (for buffer composition see in vitro kinase assay section). The reactions labelled with '(ATP)' were additionally supplemented with 100 µM ATP. All samples were incubated at RT for 20 min and stopped by adding SDS loading dye and heating at 95°C for 5 min. For the assay depicted in *Figure 3—figure supplement 3*, 1250 ng of OST1 and 500 ng of CPK6 were used.

These samples as well as the samples described in the 'whole plant protein extraction' section were subjected to SDS-PAGE (*Laemmli, 1970*). The 10% SDS acryl amide resolving gels were supplemented with 0.25–0.5 mg/ml Histone III-S (Sigma–Aldrich) (*Figure 3*, *Figure 3—figure supplement 2* and *Figure 3—figure supplement 3*) or without Histone III-S (*Figure 3—figure supplement 3*). After electrophoresis, the gel was washed three times with washing buffer (25 mM Tris–HCl pH 8.0, 0.5 mM DTT, 0.1 mM Na$_3$VO$_4$, 5 mM NaF, 0.5 mg/ml BSA, and 0.1% (vol/vol) Triton X-100) for 30 min each at RT, followed by two washes with renaturation buffer (25 mM Tris–HCl pH 8.0, 1 mM DTT, 0.1 mM Na$_3$VO$_4$, and 5 mM NaF) for 30 min each at RT and one wash at 4°C overnight. Then, the gels were equilibrated with reaction buffer (see in vitro kinase assay) for 30–45 min at RT and incubated in 20 ml of reaction buffer supplemented with 50 µCi [γ-$^{32}$P]-ATP (Perkin–Elmer). The reaction times were: *Figure 3A,B*, *Figure 3—figure supplement 2*, and *Figure 3—figure supplement 3*: 90 min; *Figure 3C*: 60 min; *Figure 3D*: 120 min. To stop the reactions and to remove background signals the gels were subsequently extensively washed with a solution containing 5% (vol/vol) trichloroacetic acid and 1% (vol/vol) phosphoric acid for at least six times for 15 min each. The gels were then stained with coomassie brilliant blue R-250, dried on Whatman 3MM paper, and exposed to a storage phosphor screen (Molecular Dynamics). The storage phosphor screen was scanned with a Typhoon reader (Amersham Bioscience).

For the in-gel kinase assays shown in *Figure 3A,B* and *Figure 3—figure supplement 2* all steps except the equilibration in reaction buffer and the reactions were carried out together and exactly the same way which allows the autoradiographs to be compared. The image files given by the Typhoon reader software are automatically adjusted to best display the bands with the highest intensity. Ca$^{2+}$-activated kinase signals are stronger than OST1-derived bands which renders OST1 bands hardly visible by the Typhoon reader software. To better visualize OST1 activity, the signal intensity of the ~41 kDa regions (blue box) in *Figure 3A,B* and *Figure 3—figure supplement 1* were adjusted as described in the following: The output files (.gel) of the Typhoon scanner software was opened using Fiji (*Schindelin et al., 2012*) and in order to enhance the visibility of OST1-derived bands the maximum signal was adjusted for the entire image including controls and the two gels which are compared in accordance with publication policies (http://jcb.rupress.org/content/166/1/11.full). Subsequently, the regions around 41 kDa were saved as .jpg file which was used for the preparation

of the figures. The parallel adjustment of the whole image showing both gels which are depicted in either *Figure 3A,B* or *Figure 3—figure supplement 1* allows the comparisons of band intensities within each figure. Additionally, in *Figure 3—figure supplement 1* several lanes of the same gel have been cut out indicated by the black line as explained in http://jcb.rupress.org/content/166/1/11.full.

## Quantitative bimolecular fluorescence complementation

Quantitative BiFC experiments were carried out as described in (*Waadt et al., 2008*) with changes explained in the following: BIFC vectors were altered to be USER cloning (*Nour-Eldin et al., 2006*) compatible (indicated by the 'u' addition to the vector name) as described in *Nour-Eldin et al. (2010)*. SLAC1-WT, SLAC1 S59A, SLAC1 S120A, SLAC1 S59A/S120A, CPK6, and ABI1 coding sequences were amplified using the PfuX7 polymerase (*Norholm, 2010*). SLAC1-WT, SLAC1 S59A, SLAC1 S120A, and SLAC1 S59A/S120A, were USER-cloned into pSPYCE(MR)u while CPK6 and ABI1 were inserted into pSPYNE173u, and pSPYNE(R)173u, respectively. Coding sequences of PP2AA3 (AT1G13320) and PP2AC5 (AT1G69960) were PCR-amplified using Phusion DNA polymerase (Invitrogen) and inserted SpeI/XmaI into a modified pUC19 vector including the pUBQ10 promoter (pUC-pUBQ10; [*Waadt et al., 2014*]). PP2AA3 was subcloned into the BiFC vectors pSPYCE(MR) and PP2AC5 into pSPYNE(R)173. Spinning disc confocal microscopy was performed using the following setup: Nikon Eclipse TE2000-U microscope with Nikon Plan 20×/0.40 ∞/0.17 WD; 1.3 and Plan Apo 60×/1.20 WI ∞/0.15–0.18 WD; 0.22 objectives. Attached were a CL-2000 diode pumped crystal laser (LaserPhysics Inc.), and a LS 300 Kr/Ar laser (Dynamic Laser), a Photometrics CascadeII 512 camera, a QLC-100 spinning disc (VisiTech international), and a MFC2000 z-motor (Applied Scientific Instruments). The software used to acquire the images was Metamorph (version 7.7.7.0; Molecular Devices). *Figure 4A* images depict maximum projections of z-stacks.

## Electrophysiological measurements in *Xenopus laevis* oocytes

Two electrode voltage clamp measurements in *X. leavis* oocytes were carried out as described previously (*Brandt et al., 2012*) with adjustments listed in the following. The recording solution contained 10 mM MES/Tris (pH 5.6), 1 mM $CaCl_2$, 1 mM $MgCl_2$, 2 mM KCl, 24 mM NaCl, and 70 mM Na-gluconate. Osmolality was adjusted to 220 mM using D-sorbitol. Oocytes were held at a holding potential of 0 mV, and subjected to voltage pulse from +40 mV to −120 mV, −140 mV, or −160 mV in −20 mV decrements. The amounts of injected cRNA were 10 ng except for *Figure 6—figure supplement 2A,E*. For the experiments shown in *Figure 6—figure supplement 2A,E* the amounts of injected cRNA were 5 ng of SLAC1, 0.5 ng of CPK6, and 7.5 ng of OST1.

## Acknowledgements

We thank Drs Jen Sheen (*cpk5/6/11*), Pedro L Rodriguez (*abi1-2/abi2-2/hab1-1/pp2ca-1* and *snrk2.2/2.3/ost1*), and Ping He (*cpk1/2/5/6*) for supplying mutant seeds, Dr Stephan Clemens for BiFC vector construction, Dr David G Mendoza-Cózatl for pGreenII_u vector construction and members of the Schroeder laboratory, in particular Dr Felix Hauser, for comments and discussions. This work was supported by the National Institute of Health (GM060396-ES010337), National Science Foundation (MCB1414339), and protein–protein interaction studies were supported by the Division of Chemical, Geo, and Biosciences, Office of Basic Energy Sciences, US Department of Energy (DE-FG02-03ER15449) to JIS. BB received initial support from a German academic exchange service PhD fellowship, and SM received support in part from a Japan Society for the Promotion of Science (JSPS) Postdoctoral Fellowship for Research Abroad.

## Additional information

### Funding

| Funder | Grant reference | Author |
| --- | --- | --- |
| National Institutes of Health (NIH) | GM060396-ES010337 | Julian I Schroeder |
| National Science Foundation (NSF) | MCB1414339 | Julian I Schroeder |

| Funder | Grant reference | Author |
| --- | --- | --- |
| German Academic Exchange Service | PhD Fellowship | Benjamin Brandt |
| Japan Society for the Promotion of Science (JSPS) | Postdoctoral fellowship | Shintaro Munemasa |
| U.S. Department of Energy | DE-FG02-03ER15449 | Julian I Schroeder |

The funders had no role in study design, data collection and interpretation, or the decision to submit the work for publication.

## Author contributions

BB, SM, Conception and design, Acquisition of data, Analysis and interpretation of data, Drafting or revising the article, Contributed unpublished essential data or reagents; CW, DN, TY, PGY, EP, TFB, Acquisition of data, Analysis and interpretation of data; RW, FA, Conception and design, Analysis and interpretation of data, Contributed unpublished essential data or reagents; JIS, Conception and design, Analysis and interpretation of data, Drafting or revising the article

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
