## [Decision Letter]

[Editors’ note: this article was originally rejected after discussions between the reviewers, but the authors were invited to resubmit after an appeal against the decision.]

Thank you for choosing to send your work entitled “Calcium Specificity Mechanism in Abscisic Acid Signal Transduction in *Arabidopsis* Guard Cells” for consideration at *eLife*. Your full submission has been evaluated by Detlef Weigel (Senior editor) and three peer reviewers, and the decision was reached after discussions between the reviewers. We regret to inform you that your work will not be considered further for publication.

The comments below are a synopsis of the major comments of the reviewers.

The overall concern is that this paper only incrementally changes the existing model for ABA dependent regulation of Slac1. A 2012 PNAS paper from many of the same authors concluded with a very similar model for ABA dependent regulation of Slac1, and other papers that are cited, such as [31], also support key aspects of the current model.

The main new finding is that PP2C phosphatases inhibit SLAC1 activation by directly dephosphorylating two different sites in the N terminus of SLAC1: S120, which is phosphorylated by non-Ca dependent SnRK kinases including Ost1, and S59, which is phosphorylated by Ca dependent kinases, CPKs. This adds to an earlier model where PP2C action reversed CPK dependent phosphorylation of SLAC1 and also dephosphorylated OST1, directly inactivating the kinase. The essential new idea then is that the N terminus of Slac1 serves as a coincidence detector, which is a nice addition to the current knowledge, but the reviewers considered this a ‘refinement’ of the current model rather than a significant conceptual advancement. Furthermore, the coincidence detection idea is not tested directly, for example with SLAC1 S59A, S120A single and double mutants introduced into the background of a SLAC1 loss-of-function mutant. The model predicts that such mutants would not be activated by ABA, and would not show constitutive activation in the PP2C quadruple mutant. Similarly, phospho-mimic single and double mutants in oocyte expression and in planta experiments would further enhance the understanding of ABA-signaling in guard cells.

Together with the limits of the artificial patch clamp experiments, the oocyte data do not entirely support the presented model. You concluded that both S120 and S59 have to be phosphorylated by OST1 and CPK6 synergistically to render SLAC1 active. Consequently disruption of one of the phospho-sites should result in an infunctional SLAC1 channel. However, this was not what was observed.

Due to the above mentioned limitations and the lack of certain controls and of in planta confirmation, the reviewers agreed that the manuscript is more suitable for a specialized journal. Specific comments from the three reviewers follow.

*Reviewer #1*:

1) The data in Figure 1 deal with calcium-dependent kinases *cpk5/6/11/23* in the ABA responsive regulation of calcium channels in guard cells. The focus of the text is on the role of CPK5 yet all of the studies are performed with a quadruple T-DNA insertion mutant that ablates all of these enzymes. Would it be possible to conduct a parallel analysis in mutant plant cells that lack single CPK enzymes, in particular CPK5?

2) The experiments in Figure 2 nicely complement data in the opening figure, but suffer from the same lack of specificity. Is it possible to measure the impact on calcium ion conductance in plant cells that lack individual ABL, HAB1 or PP2CA genes?

3) The data in Figure 3 are interesting, but as presented are less than convincing. The statement in the Results section “Several calcium activated bands disappeared… consistent with CPK activities associated with these molecular weights” is important to the overall hypothesis tested in this article. However, the quality of the data in Figure 3 is not sufficient to convince this reviewer of the validity of this statement.

4) The BiFC data presented in Figure 4 are interesting and well controlled. These findings suggest that SLAC1 and PP2CA directly interact, or do so in a proximity that permits formation of the fluorescent adduct. Given that the authors acknowledge that this protein-protein interaction has previously been reported there would be some merit in further emphasizing the new aspects of the binding event that are demonstrated in this new data. This pertains to the phosphorylation dependent regulation of the binding event. Can the phosphorylation sites be identified? Do phosphor-site mutants prevent protein association?

5) The data in the final figure are more mechanistic and deal with some of the concepts that I have raised above. My reading of this section suggests that most of the electrophysiological measurements were performed in *Xenopus* oocytes. Can any of these studies be conducted in mutant plants? Performing this work in a more physiologically relevant cell type would be an important addendum to this interesting study.

6) The Abstract and opening sections of the Introduction are not well articulated. Major editing would be necessary to improve the clarity of these important sections.

7) The details of the drug regimen presented in panel 4D are convoluted and hard to follow.

*Reviewer #2*:

A minor concern is that no negative control is shown for the BiFC experiments shown in Figure 4.

*Reviewer #3*:

Interpretations concerning the ABA primed Ca^2+^ sensitivity of SLAC1 phosphorylation are based on guard cell protoplast data. The consensus in the field is that protoplasts are impaired in many normal cellular functions. The cytosol of cells is strongly diluted and evidence for degradation of major guard cell functions has been published. The experiments here are carried out following a preincubation of protoplasts with ABA. The authors can therefore not exclude that the ABA-signaling complex is established and solidified by ABA incubation and therefore the dilution prior to patch clamp measurements is weaker than in the absence of ABA. This scenario would lead to similar results, but would have nothing to do with ABA-priming.

Wang et al., (2013, Plant Physiol.) showed that the ABA-insensitive ABA receptor quadruple mutant (*pyr1/pyl1/pyl2/pyl4*) displayed wild type like s-type anion current responses upon elevation of cytosolic calcium levels, contradicting the authors' hypothesis of ABA priming. Thus, calcium is sufficient for anion channel activation (without ABA priming). Moreover, reports from Chen et al., (2010, Plant J), Marten et al., (2007, Plant Physiol.), Stange et al., (2010, Plant J) and Levchenko et al., (2005, PNAS) illustrated that the ABA-signaling pathway is bifurcated: On one branch, ABA activates S-type anion channels without the need of cytosolic calcium elevations and on the other, ABA activates anion channels in a Ca^2+^ dependent manner. These findings contradict the interdependence of both pathways and the priming of the guard cell calcium sensitivity by ABA as reported by the authors here. Even more importantly, the experiments by Marten et al., (2007, Plant Physiol.), Stange et al., (2010, Plant J) and Levchenko et al., (2005, PNAS) were conducted with intact plants and a minimal invasive guard cell impalement technique, simultaneously recording the rise in cytosolic calcium and S-type anion currents. Thus in intact plants the situation seems to be different from the protoplast system.

Physiological tests, such as gas-exchange or stomatal aperture measurements are lacking completely.

The authors used CPK or PP2C quadruple mutants as well as a *snrk* triple mutant in their patch clamp and biochemical studies to show their impact in ABA-dependent stomatal closure (SLAC1 activation). Since all of these ABA signaling components not only regulate the activity of SLAC1 but also are involved in other aspects of gene regulation, it would be important to show that the equilibrium/stoichiometry between phosphatases and kinases are not disturbed in the mutants. This could be circumvented with the SLAC1 phospho-site mutants S59, S120 and a double mutant thereof, introduced into SLAC1 loss-of-function mutants.

Why did the authors include CPK11 in their studies?

In vitro kinase assays were performed to show that ABI1 and PP2CA do not inhibit CPK6 activity (although autophosphorylation was decreased). The authors should consider the activity of the recombinant proteins under the respective conditions. Why did the authors not see a reduction of CPK autophosphorylation when using the whole seedling proteins (Figure 3, wild type vs. PP2C quadruple KO)?

The authors used whole seedling protein preparations for comparisons between wild type and multiple mutants to show that PP2Cs control the activity of OST1 but not of CPKs and the interdependence of the Ca^2+^ dependent and independent pathway. The major quantity of whole seedling protoplasts consists of mesophyll cells that act differently than guard cells when exposed to ABA. Moreover, SLAC1 and OST1 represent guard cell specific proteins and are underrepresented in whole seedling preparation. This is also true for other tissue- and stimulus-specific ABA-signaling components. Thus, the authors should be very careful in drawing conclusions from experiments with samples guard cells represent the minority.

Heterologous expression in *Xenopus* oocytes was performed to finally show that OST1 and CPKs synergistically activate SLAC1 by phosphorylation. The authors claim that the phospho-site S120 is addressed by OST1 but not by CPK6/5/23 (already shown by [31]) and that S59 is specific for CPKs (shown by [12]) but is not a site for OST1.

For the reader it is difficult to understand how the author can conclude that both residues have to be phosphorylated by OST1 and CPK6 synergistically to render SLAC1 active when after disruption of one of the sites SLAC1 can still be activated. It would appear that phosphorylation of one site is sufficient (perhaps in combination with other phosphorylation sites) to render SLAC1 active (S59 for CPKs and S120 for OST1). Moreover, the fact that CPK6 alone and OST1 alone, coexpressed as BiFC constructs, are able to activate wild type SLAC1 also would seem to contradict the authors' conclusion that SLAC1 phosphorylation by OST1 primes SLAC1 and that additional phosphorylation of SLAC1 by CPK6 (upon Ca^2+^ increase) activates the anion channel.

When coexpressing wild type SLAC1 with both kinases the authors used very small quantities of CPK6 cRNA but 15x more OST1 cRNA. Only when suboptimal CPK6 concentrations were used, OST1 could enhance SLAC1 activation. This does not seem to prove kinase synergistic activation of SLAC1.

Why did the authors use 400 nM Ca^2+^ in the experiments shown in Figure 3—figure supplement 1, while for all other studies they used micromolar Ca^2+^ containing buffers?

BIFC experiments are lacking proper controls (there is no negative control, e.g. CPK6/OST1 coexpression should result in a negative BiFC).

Y2H experiments were performed to show that OST1 and CPK6 do not interact with each other. This negative result should be confirmed by alternative approaches.

[Editors’ note: what now follows is the decision letter after the authors submitted for further consideration.]

Thank you for sending your work entitled “Calcium Specificity Signaling Mechanisms in Abscisic Acid Signal Transduction in *Arabidopsis* Guard Cells” for consideration at *eLife*. Your article has been favorably evaluated by Detlef Weigel (Senior editor) and two outside reviewers.

Both reviewers agreed that you have addressed their major concerns, except for identification of the phosphorylation sites (see original point 4 of this reviewer below). Would it be possible to identify these and test the corresponding mutants?

“4) The BiFC data presented in Figure 4 is interesting and well controlled. These findings suggest that SLAC1 and PP2CA directly interact, or do so in a proximity that permits formation of the fluorescent adduct. Given that the authors acknowledge that this protein-protein interaction has previously been reported there would be some merit in further emphasizing the new aspects of the binding event that are demonstrated in this new data. This pertains to the phosphorylation dependent regulation of the binding event. Can the phosphorylation sites be identified? Do phosphor-site mutants prevent protein association?”

*Reviewer #1*:

I have read with interest the revised manuscript . The authors have done a creditable job in responding to the previous reviewers’ comments and have clearly articulated their concern with the actions of a specialized reviewer during the previous rounds of evaluation.

In general I am satisfied with the new data that has been added to the amended manuscript. I only have one suggestion.

1) I think that there would be some merit in the inclusion of aspects of the BiFC phosporylation studies that are discussed in point 4 of my original critique.

*Reviewer #2*:

I feel that the authors have responded to all the major issues we raised in the earlier review, and that the current version is acceptable for publication.

In particular addition of the in planta data on the slac1 mutants lacking either one or both phosphorylation sites (S59A and S120A) (Figure 6) is interesting and novel and adds significantly to the model presented in Figure 7.

---

## [Author Response]

[Editors’ note: the author responses to the first round of peer review follow.]

We have revised the manuscript according to the reviewers' comments and questions and have made additional advances in our in planta analyses, that further underline the major relevance of our findings. The revised manuscript provides strong in vivo evidence using genetic, biochemical and dynamic cell

signaling analyses for molecular mechanisms mediating abscisic acid-induced Ca^2+^ sensitivity priming. Our manuscript also addresses the question of how stimulus-specific Ca^2+^ signaling can be achieved in a plant cell, using abscisic acid-induced stomatal closing of plant guard cells. These findings could represent a general Ca^2+^-specificity mechanism and can advance the understanding of similar questions in plants and potentially other eukaryotic systems. In addition, our in planta data show the unexpected interdependence of the Ca^2+^-independent and Ca^2+^-dependent signaling pathway, contrary to present less-directly derived models, which will be of interest to a broad readership.

*The overall concern is that this paper only incrementally changes the existing model for ABA dependent regulation of Slac1. A 2012 PNAS paper from many of the same authors concluded with a very similar model for ABA dependent regulation of Slac1, and other papers that are cited, such as*
[31]*, also support key aspects of the current model*.

We list here in brief form some of the major new findings in our manuscript. Please note that the 2009, 2010, and 2012 papers referenced above focus mainly on analyzing reconstituted signaling proteins in *Xenopus* oocytes and in vitro*,* but our present study reveals several new and relevant findings in planta by analyzing higher order *Arabidopsis* mutants. The above reconstitution studies do not answer the question of how Ca^2+^-dependent and Ca^2+^-independent ABA signaling pathways are coordinated in guard cells and how Ca^2+^-signaling specificity is achieved in guard cell ABA-signaling.

The major findings in our manuscript are, in brief:

a) We have found a strong disruption in ABA-triggered guard cell S-type anion channel activation in *cpk* quadruple mutants and now also demonstrate that higher order knock out *CPK* mutants are strongly impaired in in vivo stomatal responses to 5 µM ABA underlining our findings as suggested by reviewers. We further provide evidence for redundancy in the pathway at high ABA concentrations.

b) We have found the quadruple knock out of PP2C protein phosphatases leads to constitutive Ca^2+^-signaling and ion channel regulation in planta, a major finding that provides a first mutant that removes the requirement for ABA-induced Ca^2+^ sensitivity priming.

c) Using *slac1* complementation lines (*slac1* mutants expressing WT SLAC1, SLAC1 S59A, S120A, or S59A/S120A under the native *SLAC1* promoter), now we provide interesting and unexpected *in vivo* evidence that both of the phospho-sites are required for guard cell ABA responses in planta. These findings will be of great interest to plant scientists.

d) Present models assume that the Ca^2+^-independent and Ca^2+^-dependent ABA signal transduction branches are independent from one another. Using higher order Ca^2+^-independent (*snrk2*) and Ca^2+^-dependent (*cpk*) protein kinase mutants we demonstrate genetically and in cell signaling analyses in *Arabidopsis* guard cells that these two pathways are interdependent, a question that has not been addressed directly in planta to our knowledge. We found that disruption of three Ca^2+^-independent SnRK2 protein kinase genes unexpectedly disrupts guard cell Ca^2+^ signaling, providing genetic evidence for interdependence of Ca^2+^-dependent and Ca^2+^-independent ABA signaling pathways. This in vivo evidence will have major impact on the field, where presently mainly two independent pathways are considered.

*The main new finding is that PP2C phosphatases inhibit SLAC1 activation by directly dephosphorylating two different sites in the N terminus of SLAC1: S120, which is phosphorylated by non-Ca dependent SnRK kinases including Ost1, and S59, which is phosphorylated by Ca dependent kinases, CPKs. This adds to an earlier model where PP2C action reversed CPK dependent phosphorylation of SLAC1 and also dephosphorylated OST1, directly inactivating the kinase. The essential new idea then is that the N terminus of Slac1 serves as a coincidence detector, which is a nice addition to the current knowledge, but the reviewers considered this a ‘refinement’ of the current model rather than a significant conceptual advancement. Furthermore, the coincidence detection idea is not tested directly, for example with SLAC1 S59A, S120A single and double mutants introduced into the background of a SLAC1 loss-of-function mutant. The model predicts that such mutants would not be activated by ABA, and would not show constitutive activation in the PP2C quadruple mutant. Similarly, phospho-mimic single and double mutants in oocyte expression and in planta experiments would further enhance the understanding of ABA-signaling in guard cells*.

To address this comment, we have generated and investigated SLAC1 mutants using the phospho-mimic approach and found that substituting S59 and/or S120 with Aspartate did not lead to constitutive active SLAC1 channel ([12] and unpublished results). To address the role of phosphorylated sites more directly in planta, we generated and analyzed *slac1* complementation lines (*slac1* mutants expressing WT SLAC1, SLAC1 S59A, S120A, or S59A/S120A under the native *SLAC1* promoter). In the revised manuscript we provide evidence that both of the phospho-sites are required for the full ABA response in vivo. These new findings also point to limitations of the *Xenopus* oocyte system (every system has its limits) and the relevance of our new in planta findings.

*Together with the limits of the artificial patch clamp experiments, the oocyte data do not entirely support the presented model. You concluded that both S120 and S59 have to be phosphorylated by OST1 and CPK6 synergistically to render SLAC1 active. Consequently disruption of one of the phospho-sites should result in an infunctional SLAC1 channel. However, this was not what was observed*.

Using *slac1* complementation lines (*slac1* mutants expressing WT SLAC1, SLAC1 S59A, S120A, or S59A/S120A under the native promoter), now we provide unexpected and exciting in planta evidence that both phospho-sites are required for the full ABA response in vivo. Independent patch clamp and stomatal movement experiments show consistent results in the revised manuscript. These new findings also point to limitations of the *Xenopus* oocyte system (rather than the in planta data) and point to the relevance of our new in planta findings.

We also address the reviewers’ comment regarding patch clamp techniques further below.

In brief, patch clamping has led to the development of many aspects of present guard cell-signaling models that have been verified using many independent approaches and patch clamp techniques are not known to generate “artificial” models (not-with-standing that every technique has some limitation), in contrast to one of the reviewer comments. This specific comment is directly addressed further below in more detail.

Reviewer #1:

*1) The data in*
Figure 1
*deal with calcium-dependent kinases* cpk5/6/11/23 *in the ABA responsive regulation of calcium channels in guard cells. The focus of the text is on the role of CPK5 yet all of the studies are performed with a quadruple T-DNA insertion mutant that ablates all of these enzymes. Would it be possible to conduct a parallel analysis in mutant plant cells that lack single CPK enzymes, in particular CPK5*?

We agree that the proposed experiments add to our study. The data in Figure 1 (A to D in the revised manuscript) focus on ABA and Ca^2+^ regulation of S-type anion channels in guard cells. As requested by the reviewer, we have analyzed ABA-regulation of S-type anion channels in *cpk5-1* mutant guard cells (Figure 1—figure supplement 1). The single *cpk5-1* mutation does not disrupt ABA activation of S-type anion channels, supporting the functional redundancy model of CPKs. We have also added new experiments investigating ABA activation of calcium channels, as requested (Figure 5 and Figure 5—figure supplement 1). These ion channels were not studied in our initial submission. These are described in more detail further below, and they further narrow the possible models for the ABA signaling cascade and thus add new relevant information that will be of interest to the reviewers. We note that we have done these Ca^2+^ channel regulation experiments in two laboratories and both show the same results.

*2) The experiments in*
Figure 2
*nicely complement data in the opening figure, but suffer from the same lack of specificity. Is it possible to measure the impact on calcium ion conductance in plant cells that lack individual ABL, HAB1 or PP2CA genes*?

Previous research has shown that the dominant active *pp2c* point mutations in *abi1-1* and *abi2-1* disrupt ABA regulation of ion channels ([74]; Murata et al., 2001). However, since then no phenotypes in ion channel regulation have been reported for recessive single *pp2c* insertion mutants, likely due to redundancy in these *PP2Cs*. In addition, except for the dominant active *abi1-1*, *abi2-1*, and *hab1G246D* mutants ([74]; Allen et al., 1999; Merlot et al., 2001; Murata et al., 2001; Yoshida et al., 2006), no strong stomatal phenotypes of single recessive *pp2c* knock out plants have been reported. These results further support the hypothesis that PP2Cs have overlapping functions in guard cells. Furthermore, reports showing that SLAC1 activation by OST1 and CPKs in *Xenopus* oocytes is inhibited by the ABI1, ABI2, and PP2CA PP2C protein phosphatases support functional overlapping roles of these *PP2Cs* in ion channel regulation in guard cells.

Our study reveals that quadruple knockout of the 4 major PP2Cs in guard cell ABA signaling results in constitutive Ca^2+^ activation of S-type anion channels in guard cells. This finding shows that PP2Cs are an important mechanism that controls Ca^2+^ sensitivity priming in guard cells and identifies a first mutant that causes constitutively primed Ca^2+^ sensitivity in guard cells. These results also correlate with in planta stomatal responses and whole plant phenotypes of higher order *pp2c* mutants that show constitutive stomatal closing. We note that whether there are specific roles of PP2Cs in particular sub-branches within guard cell ABA-signaling remains to be determined and should be the subject of future research, given previous studies showing no strong phenotypes of single recessive mutations in these PP2Cs.

*3) The data in*
Figure 3
*are interesting, but as presented are less than convincing. The statement in the Results section “Several calcium activated bands disappeared… consistent with CPK activities associated with these molecular weights” is important to the overall hypothesis tested in this article. However, the quality of the data in*
Figure 3
*is not sufficient to convince this reviewer of the validity of this statement*.

We agree with the reviewer that, given that there are 34 *CPK* genes in the *Arabidopsis* genome and these CPKs are widely expressed, visibility of these activities has limitations. To improve the visibility of these in-gel kinase activities from plant extracts, we now present a new figure showing a close up view of the bands corresponding to CPKs (MW 50-90 kDa) (Figure 3—figure supplement 1). Reduced Ca^2+^-activated protein kinase activities are visible. Additionally, we edited and added text in the manuscript to describe these findings more cautiously. To test stimulus-triggered activation of SnRK2 and CPK kinases, in-gel kinase assays from plant tissues are commonly employed. To test the ABA-activation of CPKs we pursued a similar approach as done for the ABA-activation of SnRK2 protein kinase activity as described in previous studies (65; 28; 101). Another line of evidence, that the bands observed in the in-gel kinase assays are indeed CPK derived, is the activation of protein kinases in in-gel kinase assays by elevating the free Ca^2+^-concentration in the reaction buffer which was performed in a comparable manner in [10] when analyzing pathogen responses of CPKs (Figure 1, [10] Nature).

*4) The BiFC data presented in*
Figure 4
*are interesting and well controlled. These findings suggest that SLAC1 and PP2CA directly interact, or do so in a proximity that permits formation of the fluorescent adduct. Given that the authors acknowledge that this protein-protein interaction has previously been reported there would be some merit in further emphasizing the new aspects of the binding event that are demonstrated in this new data. This pertains to the phosphorylation dependent regulation of the binding event. Can the phosphorylation sites be identified? Do phosphor-site mutants prevent protein association*?

The experiments suggested by the reviewer are interesting. We show in our manuscript that the PP2C ABI1 physically interacts with SLAC1 in plant cells (not previously reported), similar to PP2CA-SLAC1 BiFC interaction (which was previously reported by Lee et al.). ABI1 and PP2CA rapidly de-phosphorylate the N-terminus of SLAC1 when previously phosphorylated by SLAC1-activating kinases (which also requires a physical interaction). We note that these PP2Cs dephosphorylate multiple residues in the SLAC1 protein and we feel that the reviewer’s hypothesis that the PP2C-SLAC1 interaction may depend on the phosphorylation status of these two key sites (see reviewer comments further below), goes beyond the scope and findings of our manuscript. In the revised manuscript we have repeated BiFC experiments and now show negative and positive controls as requested in other reviewer comments. Furthermore, we have conducted initial analyses using BiFC with site-directed mutations at these sites, as proposed and hypothesized by the reviewer, and have not found a complete disruption of the interaction. As mentioned above, we feel that the proposed model of the reviewer goes beyond the scope of our study and it would be preferable to investigate such a hypothesis using several independent techniques in the future and by investigating additional phosphorylation sites and additional PP2Cs. If deemed important, we could add the new experimental results to the supplemental data.

Furthermore, we have now generated in planta complementation analysis lines and completed in vivo patch clamp and stomatal response analyses. Interestingly, these data show that only when both serine 59 and serine 120 are simultaneously substituted with a non-phosphorylatable alanine (SLAC1 S59A S120A), no ABA-activation of S-type anion currents and ABA-dependent stomatal closure can be detected (Figure 6 and Figure 6—figure supplement 3). This indicates that these 2 amino acids together are essential for the activation in planta. These new and relevant data are also discussed below in response to comment (5).

*5) The data in the final figure are more mechanistic and deal with some of the concepts that I have raised above. My reading of this section suggests that most of the electrophysiological measurements were performed in* Xenopus *oocytes. Can any of these studies be conducted in mutant plants? Performing this work in a more physiologically relevant cell type would be an important addendum to this interesting study*.

We thank the reviewer for suggesting these interesting experiments. As requested, we have now completed generation of SLAC1 phosphorylation site mutant lines in the *slac1* background in planta and conducted patch clamp and stomatal movement analyses using *slac1-1* complementation lines expressing SLAC1 WT, S59A, S120A or S59AS120A under the native promoter (new Figure 6 and Figure 6—figure supplement 3). These experiments have taken time to complete, but the results are unexpected and of major importance for understanding in planta functions. The data unexpectedly demonstrate that phosphorylation of either S59 or S120 (together with other phosphorylated amino acids) is sufficient for ABA activation of S-type anion channels and stomatal closing in planta. However, if both S59 and S120 are non-phosphorylatable, no ABA-activation of S-type anion currents can be observed. Furthermore, we have conducted stomatal response analyses and have obtained results that also independently show these interesting findings. Together these data provide in planta evidence that Serine 59 and Serine 120 together are key amino acids during ABA-dependent S-type anion current activation and stomatal closing in planta. We have revised the manuscript to report these interesting findings. Moreover, as discussed later these findings show that models derived mainly from *Xenopus* oocytes are simplified compared to the in planta signaling network.

*6) The Abstract and opening sections of the Introduction are not well articulated. Major editing would be necessary to improve the clarity of these important sections*.

We appreciate the reviewer’s comment. We have edited the Abstract and Introduction to improve clarity.

*7) The details of the drug regimen presented in panel 4D are convoluted and hard to follow*.

To improve, we revised the figure showing the sequence of recombinant proteins and additions to the reaction.

Reviewer #2:

*A minor concern is that no negative control is shown for the BiFC experiments shown in*
Figure 4.

We have repeated the BiFC experiments and included a negative control (Figure 4). As we report the interaction of the phosphatase ABI1 with SLAC1, we chose a PP2AC5 protein phosphatase catalytic subunit as negative control. We also include a positive control verifying that the PP2AC5 is expressed and shows a strong interaction with its regulatory PP2AA subunit binding partner.

Reviewer #3:

*Interpretations concerning the ABA primed Ca*^*2+*^
*sensitivity of SLAC1 phosphorylation are based on guard cell protoplast data. The consensus in the field is that protoplasts are impaired in many normal cellular functions. The cytosol of cells is strongly diluted and evidence for degradation of major guard cell functions has been published. The experiments here are carried out following a preincubation of protoplasts with ABA. The authors can therefore not exclude that the ABA-signaling complex is established and solidified by ABA incubation and therefore the dilution prior to patch clamp measurements is weaker than in the absence of ABA. This scenario would lead to similar results, but would have nothing to do with ABA-priming*.

Ca^2+^ sensitivity priming of S-type anion channel activation has also been found in native guard cells as was cited in our manuscript (15). Please also note that the original observations that led to the Ca^2+^ sensitivity priming hypothesis that stomatal closing stimuli enhance (prime) the Ca^2+^ sensitivity of downstream stomatal closing mechanisms were obtained in intact plant leaf epidermes from time-resolved Ca^2+^ imaging studies and stomatal response analyses with parallel intact leaf gas exchange analyses, a study in which no protoplast experiments were included (102; 100). Thus experiments in intact plants led to the original hypothesis and strongly support stimulus-induced Ca^2+^ sensitivity priming guard cells.

Patch clamp studies using protoplasts might affect some guard cell functions (as for all techniques), but this approach has led to the present models of guard cell ion channel signaling and stomatal regulation without major inaccuracies, which have been supported and confirmed through interdisciplinary studies using several independent approaches. Thus we respectfully disagree with this comment regarding the “consensus in the field”, and have not been able to find such a “consensus” in the literature. Patch clamp studies in the present study allow us to evaluate in planta channel regulation under more well-controlled and well-defined conditions than reconstitution in *Xenopus* oocytes. Moreover, all patch clamp experiments directly compare mutant responses (*cpk, snrk2* and *pp2c* mutants) with wild type responses and show strong and clear differences to the wildtype response that correlate with genotype-blinded stomatal movement analyses that do not use protoplasts here. Combined with biochemical and genetic evidence, our electrophysiological data provide strong evidence for newly recognized molecular mechanisms and a first genetic mutant that causes constitutive Ca^2+^ sensitivity priming in a plant cell.

*Wang et al., (2013, Plant Physiol.) showed that the ABA-insensitive ABA receptor quadruple mutant (*pyr1/pyl1/pyl2/pyl4*) displayed wild type like s-type anion current responses upon elevation of cytosolic calcium levels, contradicting the authors' hypothesis of ABA priming. Thus, calcium is sufficient for anion channel activation (without ABA priming). Moreover, reports from Chen et al., (2010, Plant J), Marten et al., (2007, Plant Physiol.), Stange et al., (2010, Plant J) and Levchenko et al., (2005, PNAS) illustrated that the ABA-signaling pathway is bifurcated: On one branch, ABA activates S-type anion channels without the need of cytosolic calcium elevations and on the other, ABA activates anion channels in a Ca*^*2+*^
*dependent manner. These findings contradict the interdependence of both pathways and the priming of the guard cell calcium sensitivity by ABA as reported by the authors here. Even more importantly, the experiments by Marten et al., (2007, Plant Physiol.), Stange et al., (2010, Plant J) and Levchenko et al., (2005, PNAS) were conducted with intact plants and a minimal invasive guard cell impalement technique, simultaneously recording the rise in cytosolic calcium and S-type anion currents. Thus in intact plants the situation seems to be different from the protoplast system*.

First [95] do not contradict our hypothesis. External Ca^2+^ shock evokes S-type anion channel activation and stomatal closure via a different mechanism from that of ABA as has been clearly reported in a number of studies. For example, a thylakoid membrane Ca^2+^ sensing protein, CAS (Nomura et al., 2008 Plant J; Weinl et al., 2008 New Phytol; Han et al., 2003 Nature) and Ca^2+^-regulated vacuolar cation channel TPC1 (Peiter et al., 2005 Nature) are involved in external Ca^2+^ shock-induced stomatal closure, but not ABA-induced stomatal closure. Ca^2+^ has long been reported to be able to by-pass ABA signaling. The other publications described above by the review (Marten et al., 2007; Stange et al., 2010; Levchenko et al., 2005) were not conducted on defined genetic mutants but rather wild-type guard cells in *Vicia faba* and tobacco and thus did not genetically dissect the Ca^2+^-dependent and Ca^2+^-independent pathways. The reviewer also points out a bifurcated ABA signaling model. By isolation and analyses of higher order mutants in both of these pathways, we now demonstrate here for the first time that these two previously considered independent pathways are intimately dependent on one another and cannot be explained by the present simple bifurcated model. This is one of several major advances reported in our manuscript. Thus we respectfully and strongly disagree with the above comments of the reviewer.

*Physiological tests, such as gas-exchange or stomatal aperture measurements are lacking completely*.

To address this comment and a comment of reviewer 1, we have completed stomatal movement analyses and added figure panels showing ABA-dependent stomatal aperture measurements using the *cpk* quadruple mutants (Figure 1) and impaired Ca^2+^-induced stomatal closing in *snrk2.2/snrk2.3/ost1* triple mutant epidermis (Figure 5). The obtained results from genotype-blinded experiments provide in vivo evidence of the importance of CPKs in ABA signaling in intact guard cells. Furthermore, we have also completed stomatal movement response analyses with *slac1* complementation lines expressing SLAC1 phosphorylation site mutants. These studies reveal that both the S59 and S120 phosphorylation sites need to be mutated to disrupt ABA signaling in planta (Figure 6 and Figure 6—figure supplement 3).

*The authors used CPK or PP2C quadruple mutants as well as a* snrk *triple mutant in their patch clamp and biochemical studies to show their impact in ABA-dependent stomatal closure (SLAC1 activation). Since all of these ABA signaling components not only regulate the activity of SLAC1 but also are involved in other aspects of gene regulation, it would be important to show that the equilibrium/stoichiometry between phosphatases and kinases are not disturbed in the mutants. This could be circumvented with the SLAC1 phospho-site mutants S59, S120 and a double mutant thereof, introduced into SLAC1 loss-of-function mutants*.

As described in response to Reviewer 1 (see above), we have performed patch clamp analyses and stomatal movement analyses using *slac1* loss-of-function mutants expressing SLAC1 phospho-site mutants. The new in vivo evidence expands our findings and shows that both phosphorylation sites (S59 and S120) need to be mutated in order to disrupt ABA signaling in planta. These interesting findings will influence and revise the present model derived in large part from oocyte studies.

Why did the authors include CPK11 in their studies?

We included CPK11 due to the high transcript abundance in guard cells and a reported ABA phenotype (103; 32).

*In vitro kinase assays were performed to show that ABI1 and PP2CA do not inhibit CPK6 activity (although autophosphorylation was decreased). The authors should consider the activity of the recombinant proteins under the respective conditions. Why did the authors not see a reduction of CPK autophosphorylation when using the whole seedling proteins (*Figure 3*, wild type vs. PP2C quadruple KO)*?

Here, we would like to further explain the experiments performed to avoid possible misinterpretation of the results. To test whether CPK activity is regulated, four independent experimental approaches were conducted and are reported in the manuscript:

1) In order to test whether CPK auto-phosphorylation can be removed by PP2Cs, an in vitro kinase assays were used (Figure 3—figure supplement 4). Here, the kinase was incubated in the presence of radioactive ATP to allow for auto-phosphorylation and subsequently the kinase was inhibited using an effective protein kinase inhibitor followed by the addition of the phosphatase. After that the proteins were separated by size and the radioactivity was monitored clearly showing that the phosphatases were able to remove CPK auto-phosphorylations in vitro.

2) The above experiment does not allow concluding whether removal of the auto-phosphorylated sites in the CPK regulates the protein kinase activity. To address this question we used a second approach: An in-gel protein kinase assay with recombinant proteins. Here, kinases were either incubated with or without recombinant PP2Cs (+/- *non*-radioactive ATP). After stopping the reactions the proteins were separated by size, re-natured and subsequently incubated in a reaction buffer containing radioactive ATP (Figure 3). At this point, when the reaction takes place, the phosphatases and the kinases are immobilized, separated by size and cannot directly interact anymore which makes a direct de-phosphorylation impossible. The same applies to experiment 3.

3) Here, whole plant extracts were first separated by size and afterwards the reactions were carried out. Hence, the phosphatases were not able to directly interact with the kinases and not able to de-phosphorylate the kinases anymore (Figure 3). We show in Figure 3—figure supplement 3 that in-gel CPK6 phosphorylation signals strongly decreased in a gel without the substrate Histone-III compared to gel in which the substrate is present can be seen (Figure 3—figure supplement 3). Similar results were obtained in [10] studying the role of CPKs in pathogen responses. Therefore the in-gel kinase assays shown in this manuscript represent mainly CPK trans-phosphorylation activities and not auto-phosphorylation activities. That ABA-application to seedlings did not lead to electro-mobility shifts of CPK-derived bands indicates that in planta, the ABA-regulated phosphatases do not de-phosphorylate the CPKs in an ABA-dependent manner. To further determine whether indeed the de-phosphorylation of CPK auto-phosphorylations seen in experiment 1 (Figure 3—figure supplement 4) are not inhibiting CPK kinase activities we performed experiment 4.

4) Here, the ATP-consumption as measurement of kinase activity was measured, with the result that CPK6 ATP-consumption is not inhibited in the presence of PP2Cs (Figure 3—figure supplement 5).

We hope that the above mentioned explanations help to clarify our design of the experiments. Our biochemical findings reported in the manuscript are important for models of ABA signaling. Presently the simplest model that PP2Cs can directly down-regulate CPK activities is considered in the field, similar to SnRK2 regulation. However, this simple model had not yet been investigated directly until the above experiments. Here we show using multiple independent biochemical approaches that this simple model is not supported as the mechanism mediating ABA signaling. Moreover, our positive controls with the OST1 protein kinase show that the methods used are reliable. In the Discussion we also discuss why the more elaborate model that we uncover here may be very important for plant Ca^2+^ signaling.

*The authors used whole seedling protein preparations for comparisons between wild type and multiple mutants to show that PP2Cs control the activity of OST1 but not of CPKs and the interdependence of the Ca*^*2+*^
*dependent and independent pathway. The major quantity of whole seedling protoplasts consists of mesophyll cells that act differently than guard cells when exposed to ABA. Moreover, SLAC1 and OST1 represent guard cell specific proteins and are underrepresented in whole seedling preparation. This is also true for other tissue- and stimulus-specific ABA-signaling components. Thus, the authors should be very careful in drawing conclusions from experiments with samples guard cells represent the minority*.

Except for SLAC1 the ABA signal transduction module PYR/PP2C/SnRK2s as well as CPKs are not only found in guard cells (*Arabidopsis* eFP Browser, http://bar.utoronto.ca/efp/cgi-bin/efpWeb.cgi). There is ample literature that PYR/PYL, PP2Cs, OST1 and CPKs are involved several ABA-dependent plant responses outside of the guard cell such as seed germination and root growth. The seedling in gel kinase assays were designed and aimed to determine whether ABA-dependent PP2C regulation of CPKs is a general regulatory mechanism as it is shown for OST1.

*Heterologous expression in* Xenopus *oocytes was performed to finally show that OST1 and CPKs synergistically activate SLAC1 by phosphorylation. The authors claim that the phospho-site S120 is addressed by OST1 but not by CPK6/5/23 (already shown by*
[31]*) and that S59 is specific for CPKs (shown by*
[12]*) but is not a site for OST1*.

*For the reader it is difficult to understand how the author can conclude that both residues have to be phosphorylated by OST1 and CPK6 synergistically to render SLAC1 active when after disruption of one of the sites SLAC1 can still be activated. It would appear that phosphorylation of one site is sufficient (perhaps in combination with other phosphorylation sites) to render SLAC1 active (S59 for CPKs and S120 for OST1). Moreover, the fact that CPK6 alone and OST1 alone, coexpressed as BiFC constructs, are able to activate wild type SLAC1 also would seem to contradict the authors' conclusion that SLAC1 phosphorylation by OST1 primes SLAC1 and that additional phosphorylation of SLAC1 by CPK6 (upon Ca*^*2+*^
*increase) activates the anion channel*.

We agree with points in this comment. As described earlier we have now generated and characterized in planta point-mutated SLAC1 complementation lines (Figure 6 and Figure 6—figure supplement 3) by performing patch clamp experiments and stomatal movement experiments. We have found that both S59A and S120A mutations together are required to disrupt ABA activation of S-type anion channels and stomatal closing in guard cells. These data also show the value of such experiments in guard cells compared to *Xenopus* oocytes. We revised the text of the manuscript. In addition to these guard cell and stomatal movement data, we have pursued further *Xenopus* oocyte data that show that the additive effect is dependent on protein kinase activity of intact OST1 (non-split-YFP tagged) (Figure 6—figure supplement 2). Our new in planta data also show some limitations of interpreting only oocyte data, even though oocytes allow important testing of models, as done in previous studies. We suspect that expressing high concentrations of protein kinases in oocytes may play a role (rather than too low protein kinase activities in oocytes as proposed by the reviewer). Meaning that expression of a high concentration of a protein kinase may well be able to activate the heterologous target protein via mechanisms that are not active in planta. This may not be surprising to some readers, but this limitation of oocyte experiments has not previously been demonstrated for the ABA signaling pathway and will add additional helpful information for the community. Thus in the present manuscript almost all electrophysiological studies are performed in guard cells and with direct comparisons of WT and higher order *cpk* , *snrk2* and *pp2c* mutants and parallel stomatal movement response analyses. In response to the reviewer comment and based on our new data we have re-worded the interpretation of the results in the manuscript and avoid over-interpreting data from oocyte experiments.

*When coexpressing wild type SLAC1 with both kinases the authors used very small quantities of CPK6 cRNA but 15x more OST1 cRNA. Only when suboptimal CPK6 concentrations were used, OST1 could enhance SLAC1 activation. This does not seem to prove kinase synergistic activation of SLAC1*.

We carefully reworded the manuscript and took the latest results into consideration and have removed the term “synergistic” in this context to address this comment. We believe that expressing high concentrations of active protein kinases in oocytes may be more subject to possible problems rather than the low kinase concentrations used in these experiments. Note also that recombinant CPK6 protein kinase activity is known to be approximately 100-fold higher than recombinant OST1 protein kinase activity (12) and therefore higher OST1 mRNA levels than CPK6 levels were used). In either case, oocytes are a simplification of the *in vivo* system, as our manuscript demonstrates. Our manuscript investigates only specific linked aspects in oocytes, with all other electrophysiological experiments focusing on guard cells. Including such experiments has value for readers, given the relevance of oocyte studies to date. Our new experiments in guard cells also provide genetic evidence that oocyte-derived models, though powerful, have limits.

*Why did the authors use 400 nM Ca*^*2+*^
*in the experiments shown in*
Figure 3—figure supplement 1*, while for all other studies they used micromolar Ca*^*2+*^
*containing buffers*?

We used this intermediate free Ca^2+^ concentration (Figure 3—figure supplement 2), in addition to 0.15 and 3 uM free Ca^2+^(Figure 3), as it has been reported that 400 nM of Ca^2+^ is enough to activate the CPKs in in vitro kinase reactions (Boudsocq et al., 2012; Laanemets et al., 2013). We thus report in-gel protein kinase data from plant extracts at three free Ca^2+^ concentrations.

*BIFC experiments are lacking proper controls (there is no negative control, e.g. CPK6/OST1 coexpression should result in a negative BiFC)*.

We repeated the BiFC experiments and included a negative control (Figure 4). As we show here the interaction of the phosphatase ABI1 with SLAC1 we chose a PP2A protein phosphatase catalytic subunit as negative control. We have also added a positive control for PP2A subunit interactions.

[Editors’ note: the author responses to the re-review follow.]

*Both reviewers agreed that you have addressed their major concerns, except for identification of the phosphorylation sites (see original point 4 of this reviewer below). Would it be possible to identify these and test the corresponding mutants*?

*“4) The BiFC data presented in*
Figure 4
*is interesting and well controlled. These findings suggest that SLAC1 and PP2CA directly interact, or do so in a proximity that permits formation of the fluorescent adduct. Given that the authors acknowledge that this protein-protein interaction has previously been reported there would be some merit in further emphasizing the new aspects of the binding event that are demonstrated in this new data. This pertains to the phosphorylation dependent regulation of the binding event. Can the phosphorylation sites be identified? Do phosphor-site mutants prevent protein association*?*”*

Reviewer #1:

*I have read with interest the revised manuscript. The authors have done a creditable job in responding to the previous reviewers’ comments and have clearly articulated their concern with the actions of a specialized reviewer during the previous rounds of evaluation*.

*In general I am satisfied with the new data that has been added to the amended manuscript. I only have one suggestion*.

*1) I think that there would be some merit in the inclusion of aspects of the BiFC phosporylation studies that are discussed in point 4 of my original critique*.

Several phosphorylation sites in the SLAC1 N-terminus have been identified to be phosphorylated by SLAC1 activating protein kinases in vitro. Among them, SLAC1 Serine 59 and Serine 120 have been identified to be important for kinase dependent activation in *Xenopus* oocytes to date. However, to our knowledge, significance of the other SLAC1 phosphorylation sites has not been analyzed in vivo. In this manuscript we demonstrate that in planta both Serine 59 and Serine 120 of SLAC1 are simultaneously required for intact ABA-dependent stomatal responses. We have acquired and added additional data as requested, that probe the interaction of CPK6 and ABI1 with the SLAC1 S59A, S120A, or S59A/S120A point mutants in BiFC interaction assays (Figure 6—figure supplement 5). Also, the data in Figure 4 were also updated based on the addition of new experiments (but do not affect the interpretation of Figure 4). We added text to discuss the new results.